



# Optimizing a dynamic fossil fuel CO$_2$ emission model with CTDAS (v1.0) for an urban area using atmospheric observations of CO$_2$, CO, NO$_x$, and SO$_2$

Ingrid Super[1,2], Hugo A.C. Denier van der Gon[1], Michiel K. van der Molen[2], Stijn N.C. Dellaert[1], Wouter Peters[2,3]

[1]Department of Climate, Air and Sustainability, TNO, P.O. Box 80015, 3508 TA Utrecht, Netherlands
[2]Meteorology and Air Quality Group, Wageningen University, P.O. Box 47, 6700 AA Wageningen, Netherlands
[3]Centre for Isotope Research, Energy and Sustainability Research Institute Groningen, University of Groningen, Nijenborgh 4, 9747 AG Groningen, Netherlands

*Correspondence to*: Ingrid Super (ingrid.super@tno.nl)





**Abstract.** We present a modelling framework for fossil fuel $CO_2$ emissions in an urban environment, which allows
constraints from emission inventories to be combined with atmospheric observations of $CO_2$ and its co-emitted
species CO, NOx, and $SO_2$. Rather than a static assignment of average emission rates to each unit-area of the urban
domain, the fossil fuel emissions we use are dynamic: they vary in time and space in relation to data that describe
or approximate the activity within a sector, such as traffic density, power demand, 2m temperature (as proxy for
heating demand), and sunlight and wind speed (as proxies for renewable energy supply). Through inverse
modelling, we optimize the relationships between these activity data and the resulting emissions of all species
within the dynamic fossil fuel emission model, based on atmospheric mole fraction observations. The advantage
of this novel approach is that the optimized parameters (emission factors and emission ratios, N=44) in this
dynamic model (a) vary much less over space and time, (b) allow a physical interpretation of mean and uncertainty,
and (c) have better defined uncertainties and covariance structure. This makes them more suited to extrapolate,
optimize, and interpret than the gridded emissions themselves. The merits of this approach are investigated using
a pseudo-observation-based ensemble Kalman filter inversion setup for the Dutch Rijnmond area at 1x1 km
resolution.
We find that the dynamic fossil fuel model approximates the gridded emissions well (annual mean differences <
2 %, hourly temporal r2 = 0.21–0.95), while reported errors on the underlying parameters allow a full covariance
structure to be created readily. Propagating this error structure into atmospheric mole fractions shows a strong
dominance of a few large sectors and a few dominant uncertainties, most notably the emission ratios of the various
gases considered. If these are either sufficiently well-known a-priori, or well-constrained from a dense observation
network, we find that including observations of co-emitted species improves our ability to estimate emissions per
sector relative to using $CO_2$ mole fractions only. Nevertheless, the total $CO_2$ emissions can be well-constrained
with $CO_2$ as only tracer in the inversion. Because some sectors are sampled only sparsely over a day, we find that
propagating solutions from day-to-day leads to largest uncertainty reduction and smallest $CO_2$ residuals over the
14 consecutive days considered. Although we can technically estimate the temporal distribution of some emission
categories like shipping separate from their total magnitude, the controlling parameters are difficult to distinguish.
Overall, we conclude that our new system looks promising for application in verification studies, provided that
reliable urban atmospheric transport fields and reasonable a-priori emission ratios for $CO_2$ and its co-emitted
species can be produced.



## 1 Introduction

Within the 2015 Paris Agreement, 195 nations agreed with a climate action plan in which each nation sets its own targets for carbon emission reductions and reports all efforts regularly to the UNFCCC (UNFCCC, 2015). An important role in reaching emission reduction targets is laid out for cities, which emit a large portion of the global fossil fuel $CO_2$ emissions (about 70 % according to the International Energy Agency (IEA, 2008)). The Paris Agreement also states that parties should strengthen their cooperation, also with respect to the sharing of information and good practices. Within this context it becomes increasingly important to map fossil fuel emissions and to quantify emission trends, both at the country- and city-level.

Most country-level greenhouse gas emission estimates reported to the UNFCCC are currently based on yearly fuel consumption data (bottom-up method), and are often spatiotemporally disaggregated using activity data and proxies to create spatially explicit emission inventories (Kuenen et al., 2014; Hutchins et al., 2017). Although the yearly national estimates are reasonably accurate (estimated uncertainty for developed countries is less than 8 % for $CO_2$ (Monni et al., 2004; Fauser et al., 2011; Andres et al., 2014)), their uncertainty increases rapidly when disaggregating them towards finer spatiotemporal resolutions (Ciais et al., 2010; Nassar et al., 2013; Andres et al., 2016). A method to improve emission estimates is using transport models in combination with independent observations of atmospheric mole fractions (Palmer et al., 2018), called data assimilation (DA) or inverse modelling (a top-down method). Recently, efforts have been made to apply DA techniques to the urban environment (McKain et al., 2012; Brioude et al., 2013; Lauvaux et al., 2013; Bréon et al., 2015; Boon et al., 2016; Lauvaux et al., 2016; Staufer et al., 2016; Brophy et al., 2018), but several challenges and unexploited opportunities remain.

First, urban DA studies have tried to constrain the total fossil fuel flux to validate bottom-up $CO_2$ inventories, often without considering the underlying emission process that caused the mismatch between observed and modelled concentrations. As one of very few exceptions, Lauvaux et al. (2013) used the $CO:CO_2$ concentration ratio to conclude that the emission reduction in Davos during the World Economic Forum 2012 was likely related to reduced traffic emissions, but without a quantification. However, emission reduction policies usually target specific source sectors. Therefore, an increase in fossil fuel emissions from one source sector can cause the total $CO_2$ emissions to appear stable, although a policy targeting another source sector can be effective in itself. To monitor the effect of each measure independently it becomes essential to attribute changes in the total $CO_2$ emissions to these policies and thus to specific source sectors. It is, therefore, not sufficient to constrain the total $CO_2$ flux, but we need to differentiate the total $CO_2$ signal into signals from the different source sectors. One way to accomplish this is by using additional measurements of co-emitted species and isotopes. Such measurements have previously been used in modelling studies to differentiate between biogenic and anthropogenic emissions or between fuel types (Djuricin et al., 2010; LaFranchi et al., 2013; Lopez et al., 2013; Turnbull et al., 2015; Fischer et al., 2017; Super et al., 2017b; Brophy et al., 2018; Graven et al., 2018), but also to separate between different fossil fuel sources (Lindenmaier et al., 2014; Super et al., 2017a; Nathan et al., 2018).

Second, for urban DA the fine scales (less than 1km and less than 1 hour) need to be resolved, therefore putting a higher demand on the atmospheric transport models. For example, Boon et al. (2016) mentioned that sources with a small spatial extent (point sources) are not correctly represented on a 2x2 $km^2$ grid, while these sources have a significant impact on the locally observed mole fractions. Concurrently, we have previously shown that a plume model improves the representation of sources with a limited spatial extent. Moreover, we found that the description



of short-term variations in the wind direction by the Eulerian WRF model in the vicinity of an urban area is poor
(Super et al., 2017a).
Third, the prior emissions also need to have a higher resolution for urban-scale studies to resolve the dominant
spatiotemporal variations. Previous studies have often used high-resolution emission maps developed specifically
for that region, using local data as much as possible (Zhou and Gurney, 2011; Bréon et al., 2015; Boon et al., 2016;
Lauvaux et al., 2016; Rao et al., 2017; Gurney et al., 2019). Yet such emission maps are only available for a few
data-rich regions. For other regions, continental or global emission maps (such as MACC or EDGAR) can be used
if downscaling is applied to reach the high resolution required for urban-scale inversions. For example, the
temporal downscaling can be done using typical daily, weekly and monthly profiles for each source sector (Denier
van der Gon et al., 2011), which are based on activity data (e.g. traffic counts) averaged over several years and/or
a large region. Spatial downscaling often involves proxies like population density. This spatiotemporal
downscaling introduces a large additional uncertainty due to uncertainties in the proxies. For example, Hogue et
al. (2016) have found an uncertainty of 150 % in the 1x1 ° fossil fuel $CO_2$ emissions for the US, whereas Ciais et
al. (2010) estimated the uncertainty of regional European emissions at 100 km resolution to be about 50 %.
Quantification of the uncertainty at an even higher resolution for urban applications has so far been limited (Andres
et al., 2016) (Super et al., 2019), also for most local inventories, while a correct definition of the prior error
covariance matrix for an inversion is important to get reliable output (Chevallier et al., 2006; Boschetti et al.,
2018). This currently complicates the application of DA studies to urban areas.
Here, we describe the development of an urban-scale DA framework (based on the CarbonTracker Data
Assimilation Shell (CTDAS)) which uses a dynamic fossil fuel emission model as a prior and optimizes the
parameters of this model. The dynamic fossil fuel emission model uses a wide range of (statistical) data to calculate
$CO_2$ emissions per source sector at high spatiotemporal resolution (1x1 km2 and hourly). The emission model is
dynamic in the sense that its formulation allows emissions to change as a function of rapidly varying conditions
in the emission landscape, such as the outside temperature, the traffic density, or availability of wind and solar
radiation for sustainable power generation. Using such information enables the calculation of dynamic emissions
in near real-time, as opposed to the construction of a static emission map based on statistical downscaling.
Moreover, the emission model can supply spatiotemporal emission uncertainties and error correlations between
source sectors, based on the estimated uncertainty of its model parameters. Since many of these parameters are
also used in bottom-up accounting of emissions, their uncertainty is often better established than the uncertainty
in the total emissions themselves. Finally, we use the dynamic emission model to calculate emissions of other co-
emitted species (CO, $NO_x$ and $SO_2$) from the $CO_2$ emissions using source sector specific emission ratios. These
co-emitted species are included in the DA system to facilitate source attribution, which is possible due to the
distinct emission ratios of different source sectors. The overall aim of this study is to explore how our dynamic
fossil fuel emission model and additional tracers can be used to overcome the known limitations in anthropogenic
$CO_2$ inverse modelling described above. The research questions are:
1. Can our dynamic fossil fuel emission model represent the spatiotemporal structure of a high-resolution
emission inventory, and what does it add to that on small scales?
2. Is the addition of co-emitted species beneficial for the attribution of $CO_2$ signals to specific source sectors,
and which observations help most in that effort?



3.    Does the prior error covariance structure that we build with the dynamic emissions model help the

119         optimization, and what can we learn from the posterior error covariance estimate?

In the inverse part of this study we use observing system simulation experiments (OSSEs, experiments using
pseudo-observations), applied to the urban-industrial complex of Rotterdam (Netherlands). This choice allows us
to test our new approach, while with real observations the errors in non-fossil and background fluxes, model
structure, and model transport will likely dominate the results (Tolk et al., 2008; Super et al., 2017a; He et al.,
2018) and reduce the ability to evaluate the methodology. First, we give an overview of the dynamic fossil fuel
emission model and demonstrate its applicability to the domain, followed by an introduction to the DA system
components and the model settings. Then we discuss the different experiments in which we start with the
comparison of different network configurations, one with only $CO_2$ and one including co-emitted species to
examine the ability to attribute $CO_2$ emissions to specific source sectors, and different state vectors. Another
experiment is used to examine the importance of propagating posterior parameters values and covariances. Finally,
we address the effect of cross-correlations.

## 2 Methods

### 2.1 The dynamic emission model

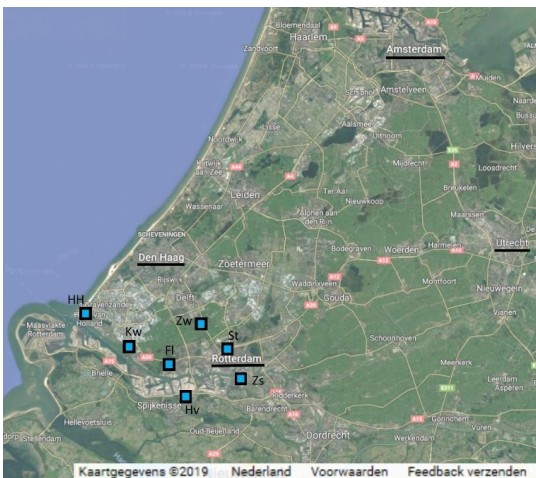

**Figure 1. Map of the domain covered (Randstad area, the Netherlands) within this study, including major cities Amsterdam, Rotterdam, The Hague, and Utrecht (underlined). The squares show the locations of the measurement sites within the urban network configuration. The area of this domain is approximately 77x88km. Source: © Google Maps.**

Although generally applicable, the dynamic emission model is initially developed for the Netherlands and focused
on Rotterdam (Fig. 1). This is one of the major cities in the Netherlands (about 625,000 inhabitants) with the
largest sea port of Europe to its west. It is located in a larger urbanized area (Randstad, about 7 million inhabitants)
with The Hague, Amsterdam and Utrecht being other major cities. A large area to the southwest of The Hague is
covered with glasshouses. The Rotterdam area is characterized by a complex mixture of residential and industrial
activities and therefore we distinguish five source sectors and a total of ten sub-sectors to construct its emissions
(see Table 1). Note that, for simplicity, only the largest source sectors are included, which are responsible for >95





% of the $CO_2$ emissions in the area. The main goal is to get a reasonable first estimate of the emission landscape
using readily available data.
**Table 1. Overview of source sectors and subsectors distinguished in the dynamic emission model, including their short**
**name used in the figures, whether they are represented as point or area sources, and their approximate contribution to**
**the total $CO_2$ emission in Rotterdam. Crosses indicate which emission factors (EF), and tracer ratios of CO, NOx or**
**SO2 (Rco, RNOx, RSO2) are part of the state vector and circles indicate whether they are also part of the short state**
**vector (see Sect. 2.3).**

| Source sector | Subsector | Short name | Source type | Contribution | EF | $R_{CO}$ | $R_{NOx}$ | $R_{SO2}$ |
|---|---|---|---|---|---|---|---|---|
| Power plants | Gas-fired power plants | 1A | Point | 37 % | XO | X | X | |
| | Coal-fired power plants | 1B | | | XO | X | X | X |
| Non-industrial combustion | Households | 2A | Area | 15 % | XO | XO | X | X |
| | Glasshouses | 2B | | | XO | X | X | |
| Industry | | 3 | Point | 39 % | XO | XO | XO | XO |
| Road traffic | Cars | 7A | Area | 6 % | XO | XO | XO | |
| | Heavy duty vehicles | 7B | | | XO | XO | XO | |
| Shipping | Ocean shipping | 8A | Area | 3 % | XO | X | XO | XO |
| | Inland shipping | 8B | | | XO | X | XO | XO |
| | Recreational shipping | 8C | | | | | | |


The emissions are calculated in four steps. First, the yearly, national emission is calculated per sector using
reported annual activity data and $CO_2$ emission factors. Second, we apply temporal disaggregation to hourly
emissions using time profiles based on a combination of default temporal profiles, and environmental conditions.
Third, we downscale the national totals to $1 \times 1$ km$^2$ resolution using statistical data, such as population density.
Finally, our approach also allows uncertainties to be described in detail based on parameters in Eq. (2).
**2.1.1 Step 1: Sectorial total emission calculations**
Total annual emissions ($F_X$ in kg yr$^{-1}$) per sector and species (X=$CO_2$, CO, NO$_x$, SO$_2$) are calculated as a function
of the economic activity and an emission factor (adapted from Raupach et al. (2007)):
$$F_X = A \left(\frac{E}{A}\right) \left(\frac{F}{E}\right) R_X \qquad (1)$$
where $A$ is the amount of activity, such as vehicle kilometres driven or generated power, and $E$ is the primary
energy consumption (petajoule (PJ)). $R_X$ is the emission ratio needed to calculate emissions of co-emitted species
$X$ from the $CO_2$ emissions, which is specific for each economic sector ($R_{CO_2}$ is always 1, others are illustrated in
Fig. 2). In this equation the term $F/E$ is the emission factor (EF), i.e. the amount of $CO_2$ emitted per amount of
energy consumed. The term $E/A$ can be seen as a measure of energy efficiency, in which technological
development plays an important role (Nakicenovic et al., 2000).





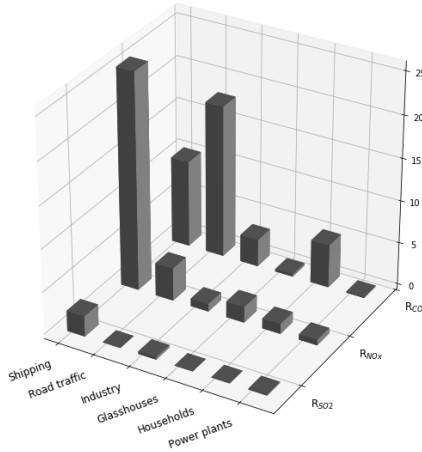

**Figure 2. Emission ratios of CO:CO₂ ($R_{CO}$), NO$_x$:CO₂ ($R_{NOx}$) and SO₂:CO₂ ($R_{SO2}$) for specific source sectors based on the Dutch Pollution Release and Transfer Register (Netherlands PRTR, 2014). Units are in ppb ppm$^{-1}$. A value of 10 on the y-axis thus implies that for each 1000 moles of CO₂, 10 moles of the auxiliary tracer is emitted.**

The information needed in Eq. (1) comes from various inventories and national information sources. For example, changes in annual activity can be approximated based on national statistics such as the GDP (Gross Domestic Product), which can be a proxy for industrial activity. Or A can be based on environmental data such as the annual degree day sum based on the outside temperature, as proxy for the need for household heating in a particular year. The second term in Eq. (1) ($E/A$, the energy efficiency) can be estimated from energy consumption statistics, such as available from the International Energy Agency. Note that this term can show a large trend in case of technological development. The last terms in Eq. (1) ($F/E$ and $R_x$, the emission factors) are the most uncertain ones, because the emission factor is dependent on the fuel mix and the energy efficiency, which itself can vary with environmental conditions (e.g. a cold engine on a winter day burns less efficiently). It can therefore differ significantly between countries. Emission factor values that are generally valid can be gathered from the Intergovernmental Panel on Climate Change (IPCC) or the European Environmental Agency (EEA), while country-specific values are typically less easily accessible. For our study area, we have access to both EEA data, as well as to Netherlands-specific numbers and even to Rijnmond-specific values (PRTR). See Appendix A for a full overview of the data used.

**2.1.2 Step 2: Temporal profiles and parameterizing activity**

The second step is to disaggregate the yearly emissions to hourly emissions by calculating time profiles, such that Eq. (1) becomes "dynamic":

$$F_{X,t} = A \left(\frac{E}{A}\right)\left(\frac{F}{E}\right) R_X T_t \tag{2}$$

where $T_t$ is the hourly time factor. Averaged over a year the value of $T_t$ is 1.0, so that it only alters the temporal evolution and not the total emissions. Energy use is often specifically linked to an activity ($A$ in Eq. (1) and Eq. (2)) on which temporal information is more readily available than on the resulting emissions. Therefore, $T_t$ can be calculated in two ways: 1) by directly using temporally explicit activity data or 2) by parameterizing temporal variations from environmental and/or economic conditions. When activity data is available the first option is

preferable. However, in data-sparse regions the second option might be necessary to implement, which is still an
improvement compared to long-term average profiles as commonly used as we will discuss next for several sectors
represented in our dynamic emission model.
Non-industrial combustion is dominated by households' natural gas consumption to heat houses, for cooking, and
for warm water supply. A Dutch energy provider has a dataset publicly available from about 80 smart meters for
the year 2013 with hourly gas consumption (Liander, 2018). It clearly shows a seasonal cycle, but also more small-
term variations (daily data are shown in Fig. 3). We also see higher gas consumption in the beginning of the year,
where the first three months of 2013 had some long, cold spells.
The use of energy for household heating is connected to the outside temperature. Previous studies have therefore
used the concept of heating degree days to describe the temporal variability in emissions from households (Mues
et al., 2014; Terrenoire et al., 2015). This concept assumes that heating only takes place below a certain temperature
threshold (here 18°C) and the hourly time factor can be defined as:
$T_t = H/\bar{D}$ (3)
where $H$ is the heating degree day factor ($H = max(291.15 - \overline{T_{2m}}, 0)$) based on the daily mean outside temperature
at 2 m. $\bar{D}$ is the yearly average heating degree day ($\bar{D} = \frac{1}{N}\sum_{j=1}^{N} H$). However, gas consumption related to warm
water supply and cooking is largely independent of the outside temperature and therefore a constant offset is
included in the heating degree day factor:
$H_f = H + f \cdot \bar{D}$ (4)
where $f$ is the constant offset. We assumed an offset of 20 %, similar to Mues et al. (2014). The time factor can
now be defined as:
$T_t = H_f / \overline{D_f}$ (5)
where the average heating degree day accounted for the constant offset $\overline{D_f} = (1+f)\bar{D}$.
We compared the heating degree day method with gas consumption data on a daily basis (Fig. 3). The degree day
function follows the gas consumption data very well, including the higher consumption at the start of the year,
reaching an $R^2$ of 0.90 (N=365). The gas consumption of consumers also has a diurnal pattern with peaks in the
early morning and late afternoon. Therefore, a diurnal profile can be estimated based on typical working hours.
For hourly data $R^2$ is 0.80 (N=8760, not shown).

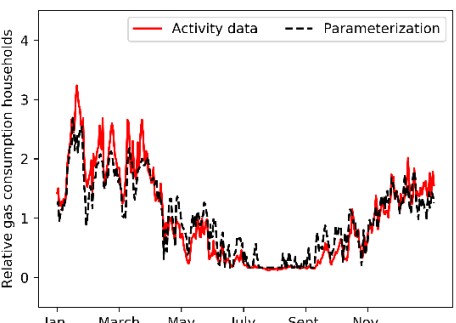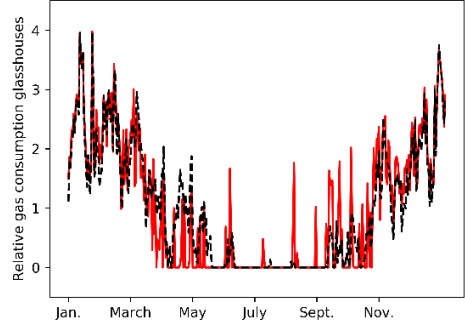

**Figure 3. Daily time profiles for households (left) and glasshouses (right). Solid red lines are based on true activity data,**
**whereas dashed black lines are parameterizations based on the degree day function.**



For the energy consumption of glasshouses there is no true activity data available. Instead, we use modelled daily
energy consumption for a typical Dutch glasshouse cultivating tomatoes (courtesy of Bas Knoll, TNO) as the
'truth' (activity data). This time profile is calculated for typical meteorological conditions, such that the order of
magnitude and the peaks are representative for an average year. There is almost no energy consumption during the
summer, which indicates that there is no constant offset. So, we use Eq. (3) to determine the emission factor.
Moreover, we use a lower temperature threshold of 15 °C to get a better fit with the observed energy consumption.
The estimated function compares well with the activity data (Fig. 3) with an $R^2$ of 0.85 (N=365).
The diurnal cycle of glasshouse emissions is likely to be different from that of household emissions. Yet we lack
data to establish a diurnal cycle. We therefore use the same diurnal profile as for households, although this is likely
to be incorrect.
Power plants can use different fuels such as hard coal, natural gas or biomass. In the Netherlands coal-fired and
gas-fired power plants account for 80–85 % of the total energy production. The remainder comes mainly from
wind energy (5–6 %) and biomass burning (5–6 %). Power generation data are reported by the European Network
of Transmission System Operators for Electricity (ENTSO-E), which has detailed data available for the whole of
Europe. Coal-fired power plants are currently the main source of energy and their generation is relatively stable
compared to other sources. It does, however, show a seasonal cycle with less energy production during the summer
months. Gas-fired power plants have a larger temporal variability as they are mainly used as back-up for peak
hours, depending also on the amount of renewable energy that is available.
We use Eq. (5) to estimate the time profiles of both coal- and gas-fired power plants. Linear regression analysis
shows that the coal-fired power generation is correlated with degree days ($R^2 = 0.17$). In this case we use a large
constant offset of 80 % and a threshold of 25 °C which were chosen to best match the actual power generation
data. The offset is much larger than for households because there is always a basic energy demand from the
industry. In contrast, the gas-fired power plants are (negatively) correlated with the wind speed ($R^2 = 0.13$) and
incoming solar radiation ($R^2 = 0.10$), indicating the need for gas-fired power generation in the absence of
renewable sources. Therefore, we replace the temperature used to calculate $H_f$ in Eq. (4) with the multiplication of
wind speed and incoming solar radiation:
$$H = \max(10 - \bar{u}, 0) \cdot \max(150 - \bar{R}, 0) \tag{6}$$
where $u$ is the wind speed (m s$^{-1}$) and $R$ the incoming solar radiation (J cm$^{-2}$). Here we use a constant offset of 10
% and a threshold of 10 m s$^{-1}$ and 150 J cm$^{-2}$.
The diurnal cycles for power plants can be based on socio-economic factors. For example, the energy demand
peaks early in the morning when people get ready to go to work and at the end of the afternoon when they get
home. We find this pattern in the actual power generation data, with coal-fired power plants being less variable
during the day than gas-fired power plants. The fixed profile from the European MACC-III emission inventory
(Denier van der Gon et al., 2011; Kuenen et al., 2014) matches reasonably well with gas-fired power plant profiles,
but it is less applicable for coal-fired power plants (Fig. 4). Overall, the estimated profiles for gas-fired power
plants (hourly data) have an $R^2$ of 0.32 (N=8784) when compared to the activity data. For coal-fired power plants
this is 0.21 (N=8784).




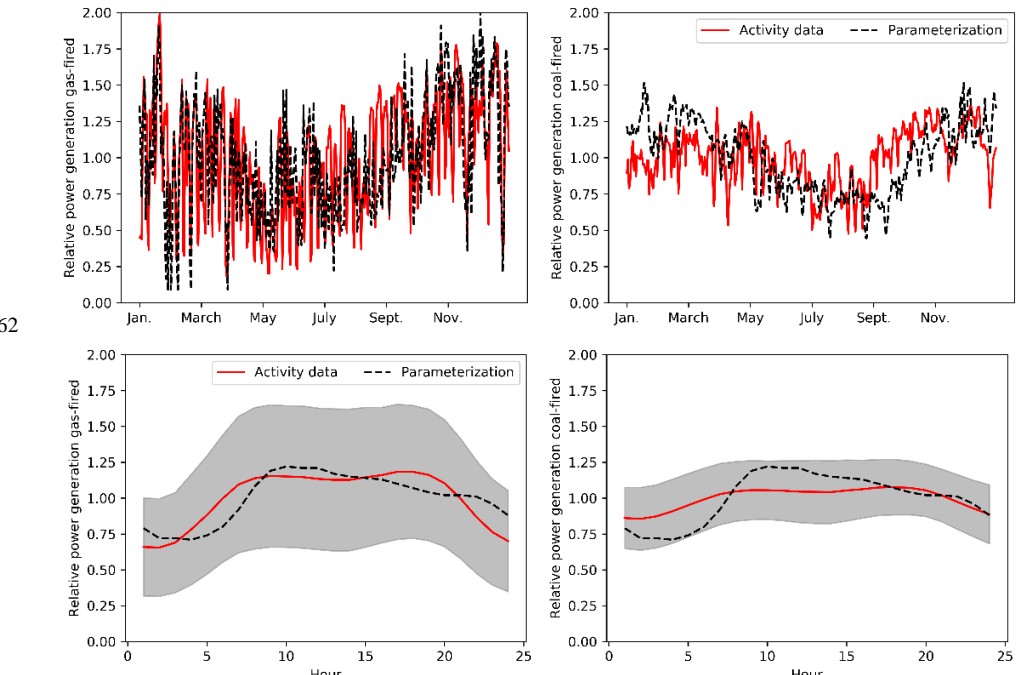


**Figure 4. (top row) Daily time profiles for gas-fired (left) and coal-fired (right) power plants. Solid red lines are based on true activity data, whereas dashed black lines are parameterizations based on observed temperature (coal) and wind speed/radiation (gas). (bottom row) Average diurnal cycle for gas-fired (left) and coal-fired (right) power plants. Solid red lines are based on true activity data, whereas dashed black lines are fixed profiles from the MACC inventory (Denier van der Gon et al., 2011; Kuenen et al., 2014). Shading gives the 1σ variability of the diurnal cycle based on activity data.**

The industrial sector consists of a wide range of activities, of which some are semi-continuous and only interrupted
by maintenance stops while others follow working hours. This makes it very difficult to predict the temporal
variability, especially for the overall sector. Since the largest $CO_2$ emissions are related to refineries and heavy
industry we will focus on these activities. We find a seasonal cycle in the reported industrial activity, with a small
decline during the summer and Christmas holidays. However, the variations are very small (max. 1 %). Therefore,
we assume constant emissions.
Road transport emissions can vary between different road and vehicle types (Mues et al., 2014), but are also
strongly dependent on environmental, socio-economic and driving conditions (such as the amount of stops, free-
flow versus stagnant conditions, and engine temperature). Traffic count data are often used to create average time
profiles for road traffic emissions, although with traffic counts we are unable to account for environmental and
driving conditions. Traffic counts for the Netherlands are made available by the Nationale Databank
Wegverkeersgegevens (NDW) and similar data is available in many developed countries. We differentiate between
two vehicle types (passenger cars + motorcycles (hereafter referred to as cars) and light duty + heavy duty vehicles
(hereafter referred to as HDV)) and three road types (highway, main road, urban road). We selected all available
locations for 2014 within or close to Rotterdam that distinguish 3-5 vehicle lengths and filtered for a minimum
data coverage of 75 %. This leaves us with 25 highway, 6 main road and 13 urban road locations. From this data
we make average time profiles (daily, weekly and monthly) per road and vehicle type, as is often done to
disaggregate road traffic emissions. Note that this method excludes any spatial variations (e.g. highways leading
towards the city vs. the beach), except for differentiating between road types.
Generally, HDV show a larger spread due to the low counts during the weekend (Fig. 5). Car counts on weekdays
show a morning and evening rush hour and they go down in between. In contrast, HDV counts peak throughout
the day and only go down after the evening rush hour. Moreover, the diurnal cycles are different during the
weekend than on weekdays. These patterns can be explained from socio-economic factors. Current time profiles
are often based on cars and are unable to correctly represent the temporal variability of HDV. This also affects the
spatial distribution of emissions and therefore we create average diurnal, weekly and seasonal profiles separately
for cars and HDV, for different road types and considering the day of the week. The comparison of true traffic
counts and averaged traffic counts results in $R^2$ values between 0.83 and 0.95 for hourly data for the whole year
(N between 2665 and 6471).

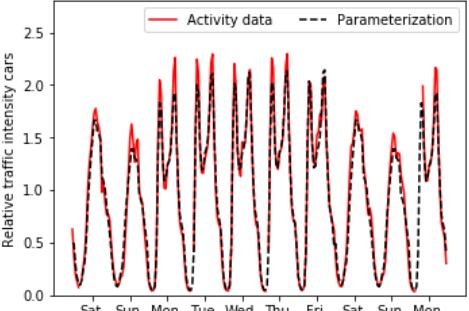 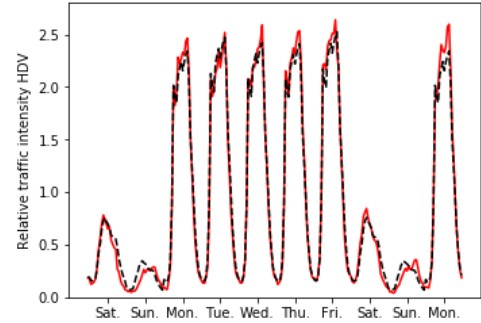


**Figure 5. Time profiles of passenger cars (left) and heavy-duty vehicles (right) road transport on highways for ten randomly chosen days in March. Solid red lines are based on true activity data, whereas dashed black lines are parameterizations based on averaged traffic counts for Rotterdam.**

Shipping emissions are dependent on the type of fuel used and whether ships apply slow-steaming. Additionally,
during loading and unloading ships still emit $CO_2$ and other pollutants, even though they are not moving. Such
information is currently not available, so instead we use information about the arrival and departure of ships in the
port of Rotterdam to make a time series of ship movements. Note that this only applies to large vessels that
transport goods and passengers and that the time profile will look quite different for recreational shipping.
However, large ships account for approximately 80 % of the total shipping emissions in the area of interest. Since
we lack information about other type of shipping movements, we will only account for large ships in the time
profiles.
We collected ship movements for one month (daily data) and an average diurnal profile. The diurnal cycle shows
a peak throughout the day, which corresponds well with the HDV road transport emission patterns on highways.
The reason for this is that HDV road transport is related to shipping movements, as HDV takes care of part of the
good transport further inland after the goods have arrived by ship. We also find a clear weekly pattern with less
ship movements during the weekend, although the decrease is less than for HDV road transport. This is likely
because large ships, such as entering the port of Rotterdam, continue travelling during the weekend. Therefore,
the weekly pattern resembles more that of car road transport on highways. Thus, we can estimate ship movements
by using the temporal profiles of HDV and cars on highways. This method is specifically tested for Rotterdam and
different patterns might be visible elsewhere. We also use HDV patterns for the seasonal variability, and final





parameterized and reported activity in this method reach an $R^2$ value of 0.89 for a period of 18 days with hourly
data (N=432) as shown in Fig 6.

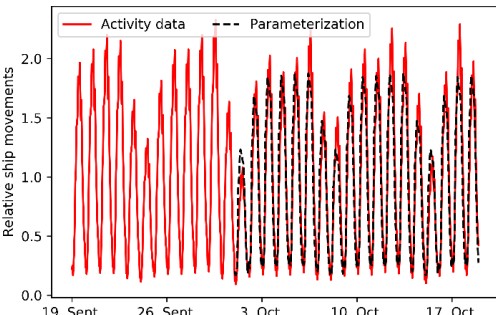

**Figure 6. Daily time profiles for shipping. Solid red line is based on true activity data, whereas dashed black line is a**
**parameterization based on traffic counts of heavy-duty vehicles (diurnal cycle) and cars (day-to-day variations) on**
**highways.**
**2.1.3 Step 3: Spatial disaggregation.**
National total sectorial emissions need to be distributed into spatially explicit emissions for our study domain. The
spatial disaggregation of emissions has received quite some attention already from inventory builders. Existing
emission inventories can be used to describe the spatial disaggregation, if available for the region at high
resolution.
If not, simple default proxies for the spatial distribution are population density and the presence of roads or
waterways (e.g. OpenStreetMap). For example, main roads and urban roads are busiest in densely populated areas
and we assume emissions on main and urban roads are correlated with population density. Highways are used for
transport between cities and therefore emissions take place outside densely populated areas as well. Nevertheless,
highway transport is usually to and from densely populated areas, such that most emissions will take place close
to cities. We can therefore relate these emissions with the population density in the area of interest (in this case
Rijnmond) relative to the rest of the country, which places the same amount of the country-level emissions in our
case study domain as the gridded inventory. Additionally, the location of large power plants or industrial plants is
often known (for example from E-PRTR (Pollutant Release and Transfer Register), which can be used directly.
Although such information allows us to possibly construct a detailed fossil fuel model in data-sparse regions in
the future, in this study we focus first on the more easily implementable and less-developed parameterization of
temporal activity in different sectors (step 2) to assess whether this approach is promising enough for future
extension.
**2.1.4 Step 4: Uncertainty analysis**
The emission model we have constructed in steps 1–3 contains several parameters per source sector: activity,
emission factor, spatial proxy and time profile. For the analysis we only consider the emission factors and time
profiles, as we assume activity data and the spatial distribution to be well-known. As input for step 1 in the dynamic
emission model we use generalized parameters which we take from the IPCC, EEA and other organizations. These
databases also provide an uncertainty range, which we use in a final step to create a covariance matrix. The
covariance matrix describes the Gaussian uncertainty of these parameters (diagonal values) and error correlations
between parameters (off-diagonal values). From the covariance matrix we create an ensemble of parameters



(N=500) that represents their joint distributions, and we use them to calculate an ensemble of emissions. In this
Monte Carlo simulation, we transform some Gaussian parameters into log-normal distributions to account for non-
negativity, or to account for distributions with a very long tail (mainly emission ratios, which can become high in
specific cases where no emission reduction measures are taken). Appendix A summarizes the used parameter
values and uncertainties (including the shape of the distributions) and shows an example of the covariance matrix.
In a final step, we select the most important parameters which are either very uncertain or have a large impact on
the total emissions. This leaves us with the 44 parameters that we optimize in a set of data assimilation experiments,
described next. In Sect. 3.1 we report uncertainties in % (1 σ) for normal distributions ($CO_2$) or as a 90 %
confidence interval (CI) for lognormal distribution (co-emitted species).
**2.2 Data assimilation to estimate fossil fuel sources**
The goal of data assimilation is to find a state at which the system is in optimal agreement with observations. In
this work, the observations we want to explore are the mole fractions of $CO_2$ and its co-emitted species while the
state of the system is the underlying spatiotemporal distribution of fossil fuel emissions. Such configurations are
sometimes referred to as "FFDAS" (fossil fuel data assimilation systems) applications, with a number of examples
in recent literature (Rayner et al., 2010; Asefi-Najafabady et al., 2014; Basu et al., 2016; Graven et al., 2018).
Given the sparsity of approaches explored so far, the dynamic emission model with its parameter driven emissions
we present here could lend itself well for application in an FFDAS, and this is what we explore through a set of
experiments with our own data assimilation methodology.
In this study we use the CarbonTracker Data Assimilation Shell (CTDAS) (v1.0) described in detail in Van der
Laan-Luijkx et al. (2017). Briefly, the CTDAS system is a flexible implementation of a square-root Ensemble
Kalman Filter (Whitaker and Hamill, 2002), which also allows lagged windows (i.e. smoothing instead of
filtering). The Ensemble Kalman Filter optimizes the cost function for unknown variables in the state vector $x$
using information from observations ($y^0$ with covariance $R$) and a prior estimate of the state vector ($x^b$ with
covariance $P$).
$$J(x) = \left(y^0 - \mathcal{H}(x)\right)^T R^{-1}\left(y^0 - \mathcal{H}(x)\right) + (x - x^b)^T P^{-1}(x - x^b) \qquad (7)$$
In this function, $\mathcal{H}$ is the observation operator that returns simulated mole fractions given the state vector. $R$ and
$P$ determine how much weight is given to the observations and prior estimate, respectively.
The optimized state vector (indicated with superscript $a$, whereas $b$ refers to the prior estimates) which minimizes
the cost function is
$$x^a = x_t{}^b + K(y_t{}^0 - \mathcal{H}(x_t{}^b)) \qquad (8)$$
and its covariance is
$$P_t{}^a = (I - KH)P_t{}^b \qquad (9)$$
Here, H is the linearized observation operator and $K$ is the Kalman gain matrix:
$$K = \left(P_t{}^b H^T\right)\left(H P_t{}^b H^T + R\right)^{-1} \qquad (10)$$





The solutions of Eq. (8) and Eq. (9) are calculated as in Peters et al. (2005) using an ensemble of 80 members. The
choice for the ensemble size was based on the typical dimensions of our inverse problem, which has N=1960
observations and M=44 unknowns for the base run.

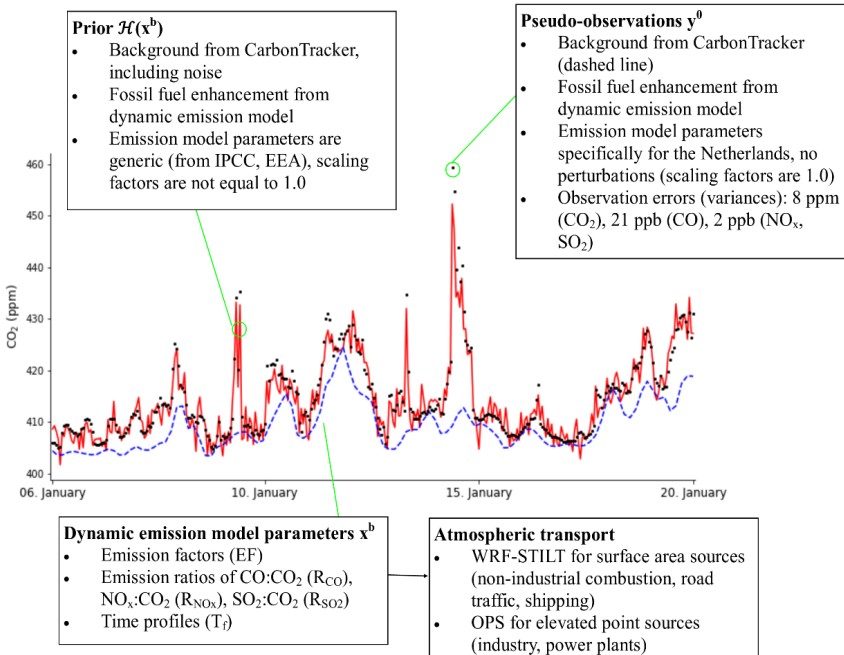

**Figure 7. Time series of pseudo-observations and prior CO₂ mole fractions and a summary of how these time series**
**were created.**
We have adapted CTDAS for smaller scale studies by replacing the typical observation operator $\mathcal{H}$, which is the
global TM5 transport model (Huijnen et al., 2010), with a combination of WRF-STILT footprints and the OPS
plume model, building on the methods described in Super et al., (2017a) and He et al. (2018). Moreover, we have
added our dynamic emission model to the observation operator so that we can sample its parameter distribution in
atmospheric mole fraction space. More details about the individual parts of this system are provided below and
are summarized in Fig. 7.
**2.2.1 Observation operator**
The observation operator translates the 44 parameters in the dynamic emission model first into emissions (through
Eq. (1) and Eq. (2)) and then into atmospheric mole fractions. The transport modelling consists of two parts. The
first part, the Weather Research and Forecasting-Stochastic Time-Inverted Lagrangian Transport (WRF-STILT,
(Nehrkorn et al., 2010) model, is used for surface emissions that are representative of large areas (i.e., not a point
source). STILT is a Lagrangian particle dispersion model that describes the footprint of a single measurement by
dispersing particles back in time (Gerbig et al., 2003; Lin et al., 2003). With this footprint the surface influence of
emissions on a single observation can be described. An advantage of this method is that it allows the pre-
calculation of linear atmospheric transport, which makes this part of the observation operator less computationally





demanding than running an ensemble of a full atmospheric transport model (like WRF with chemistry). The total
domain covered with WRF-STILT is 77 x 88 km (Fig. 1) and includes most of the Randstad.
To generate a footprint, 75 particles are released at the observation site at the start of the back-trajectory and
followed back in time. Given that the variability in hourly observations at an urban location is dominated by local
signals, we construct back-trajectories spanning 6 hours. This is based on the domain size, which could be covered
within 6 hours for typical wind speeds of 4 m s$^{-1}$. Within this time frame emissions can become well-mixed
throughout the boundary layer under normal daytime mixing conditions, such that emissions outside this range
can be represented by a boundary inflow. Footprints are generated for each hour within the back-trajectory to
account for hourly variations in the emissions. We drive STILT with meteorology from the WRF model (v3.5.1).
The WRF model was set up with two nested domains (15x15 and 3x3 km$^2$ horizontal resolution) and the STILT
footprints have a 1x1 km$^2$ resolution over the entire domain.
The second part of the transport modelling is a plume model. In a previous study we have shown that point source
(stack) emissions should be modelled with a plume model to better represent the limited dimensions of the stack
plume (Super et al., 2017a). Similarly, Vogel et al. (2013) have shown that the surface influence calculated by
STILT can lead to large model errors for stack emissions. Therefore, we include the OPS (Operational Priority
Substances, short-term version) plume model in our framework to model the transport and dispersion of stack
emissions (Van Jaarsveld, 2004; Sauter et al., 2016). OPS provides hourly concentrations at pre-defined receptor
points, which represent our measurement sites. The model keeps track of a plume trajectory, considering time-
varying transport over longer distances (e.g. changes in wind direction and dispersion). If for a time step a specific
plume affects the receptor, a Gaussian plume formulation is used to calculate the mole fraction caused by that
source based on the true travel distance along the trajectory. We drive the model with the same WRF meteorology
as STILT. Only primary meteorological variables (temperature, relative humidity, wind direction, wind speed,
precipitation, global radiation) are prescribed, secondary variables (e.g. boundary layer height, friction velocity)
are calculated by OPS itself and can differ from WRF.
Similar to the WRF-STILT model, we assume an influence time of 6 hours on our observations. However, in this
case we run the OPS model forward from -6 hours to the time of observation. We apply the OPS model only to
point source emissions within the Rijnmond area, as we found in a previous study that a plume model only has an
added value less than 10–15 km downwind from the stack (Super et al., 2017a). Point sources at more than 10–15
km from the observation site can be sufficiently represented with a Eulerian model. The OPS model input includes
detailed information about the exact stack height and heat content of the plume.
In addition to the fossil fuel contribution we also include background mole fractions for $CO_2$ and CO. $NO_x$ and
$SO_2$ are short-lived and therefore the variations in the background are relatively small compared to the fossil fuel
signals. The $CO_2$ background is taken from the 3-D mole fractions of CarbonTracker Europe (Peters et al., 2010)
and also accounts for biogenic fluxes. The resolution of these $CO_2$ fields is 1x1° and we select the grid box that is
situated over Rotterdam. The 3-hourly data are linearly interpolated to get hourly background mole fractions that
are added to the fossil fuel signals calculated by the transport models. We use the strong wintertime correlation
between $CO_2$ and CO mole fractions (r = 0.73) to calculate CO background conditions from the $CO_2$ background.
This is not very accurate, but for the purpose of this OSSE it provides us with a decent estimate of the variability
in background mole fractions.
**2.2.2 State vector**



We populated the state vector with a selection of the most important parameters of the dynamic emission model,
based on their impact on the total emission uncertainty described in the results (Sect. 3.1). However, we
hypothesize that emission model parameters that are not part of the state vector are nevertheless uncertain and may
affect the results. Therefore, we include a total of 44 scaling factors in our state vector ($x^b$), and each scaling factor
is linearly related to a parameter from the dynamic emission model. The uncertainty in these parameters
(covariance matrix $P$) is derived from the Monte Carlo simulations described in Sect. 2.1, with the spread in the
emission model parameter values provided by the same databases of the IPCC and EEA. These uncertainty values
can also be found in Appendix A.
For this study we selected an arbitrary two-week period in January 2014 (6–20 January). Note that during the
summer the importance of source sectors might be different, e.g. there will be less heating from households.
Nevertheless, this period is sufficient to test the applicability of our DA system. We loop over the 14 days in our
study period, resulting in one posterior state vector for each day. We initialize our state vector for every new day
using the posterior values and posterior uncertainties from the previous day. Because the footprints we generated
extend backwards for six hours, the state vector for each day is effectively only constrained by the observations
from that same day, and hence we did not use a Kalman-smoother approach in this work in contrast to other
CTDAS applications.
Although this is a data-rich region, we use generic values for the prior emission model parameters which we take
from the IPCC, EEA and other organisations (Appendix A). These values are typically valid for a large region
(e.g. Europe) and not necessarily the best estimate for our regional case study. The reason that we use these values
is that they can provide a first estimate of the emissions in data-scarce regions where inverse modelling might add
most to our knowledge. With this set-up we can examine how well we can constrain the true emissions starting
with this generic, and widely available, information.
One major challenge in this study is to attribute the mismatch between the observed and modelled mole fractions
to a specific sector, as a $CO_2$ observation alone provides no details on the origin of the $CO_2$. Therefore, we include
three tracers (CO, $NO_x$ and $SO_2$) that are co-emitted with $CO_2$ during fossil fuel combustion in a ratio (referred to
as $R_{CO}$, $R_{NO_x}$ and $R_{SO2}$) that is specific for each source sector (Fig. 2). Their (pseudo-)observations can inform us
about the source of the mismatch, but through their emission ratio to $CO_2$ they also constrain the magnitude of
$CO_2$ emissions in the emission model. The ratios $R_{CO}$, $R_{NO_x}$ and $R_{SO2}$ used for this conversion to $CO_2$ emissions is
not fixed: for each of the co-emitted species we included them in the state vector. This recognizes that emission
ratios are highly variable and uncertain but play an important role in source attribution.
**2.2.3 Pseudo-observations**
In this work we create observing system simulation experiments (OSSEs), which use pseudo-observations instead
of true observations. The advantage of using pseudo-observations is that we can accurately examine the abilities
of our new approach without having to account yet for (often dominant) atmospheric transport errors.
The pseudo-observations used to optimize the emission model parameters are created using the same observation
operator as described above. The dynamic emission model is used to create realistic emissions with a high
spatiotemporal resolution. Yet in contrast to the prior, we use specific local (Dutch) values for the emission model
parameters. These parameters are considered to be the truth and are therefore not scaled (scaling factors are 1.0).
We found that these local parameter values are always within the uncertainty range of the general (prior) values,



so that the true solution is part of the distribution explored within the prior. This is confirmed in an experiment
with a small model-data mismatch and no noise on the background, which reproduces the true parameters very
well (not shown).
The resulting emissions are used in combination with the background mole fractions and transport calculated by
WRF-STILT and the OPS model to create pseudo-observations at the locations shown in Fig. 1. For the pseudo-
observations the original background time series are used, whereas in the inversion random noise is added to the
background mole fractions with a standard deviation of 2 ppm for $CO_2$. We assume no contribution from biogenic
$CO_2$ to the excess $CO_2$ over the background, which means that any biogenic contribution to $CO_2$ within our
footprint is the same as in the inflow from outside our domain, thus cancelling in the subtraction of the background
$CO_2$.
One simulated time series is illustrated in Fig. 7. The monitoring network consists of seven sites that are scattered
over the city of Rotterdam and the port. All sites exist in the national $CO_2$ or air quality measurement networks,
although not all species used in the inversion are observed at all locations. We only use the daytime (12–16 h LT)
observations to constrain our emissions. This is normally done to favour well-mixed conditions when simulated
transport is more reliable, and we want to mimic this limitation. We assume all instruments have an inlet at 10m
above ground level. In reality this is lower for several sites, but during the well-mixed daytime conditions the
difference is minimal.
The covariance matrix $R$ describes the observation error. It accounts for errors related to instrumentation, but also
representativeness errors due to model transport, interpolation, and parameterization used in the dynamic emission
model. Although in principle such errors can be excluded in an OSSE, we prefer to use realistic estimates of these
errors to allow for the random errors that we applied to the prescribed boundary inflow, as well as to account for
some parameters in the emission model that are not optimized even though they contained uncertainty in the
pseudo-data creation. We base the $R$ matrix on the calculated errors and variability caused by these specific
differences, and we end up with variances of 2.5 ppm ($CO_2$), 8 ppb (CO), 3 ppb ($NO_x$) and 1 ppb ($SO_2$).
**2.3 Data Assimilation Experiments**
We perform various experiments to examine the sensitivity of the system to different set-ups and sources of error.
The experiments are discussed here, and the detailed set-up of the inversions is summarized in Table 2. The base
run is labelled "Base".
1) State vector definition: We start with a comparison of two different state vectors. For this purpose, we compare
the base run with an inversion (Short_state) which only includes the 21 most important parameters as identified in
the sensitivity analysis. This test allows us to examine the impact of erroneous, non-optimized emission model
parameters on the emission estimates. The results are discussed in Sect. 3.2.
2) Source attribution: Next we compare two monitoring network configurations which differ in the number of
tracers used. We perform an inversion with $CO_2$ as the only tracer ($CO_2$_only) and one with the full range of tracers
(Base) to assess the added value of including co-emitted species for source attribution. These tests address the
question whether co-emitted species can be used for source attribution. The results are discussed in Sect. 3.2.
3) Propagation: The third experiment is used to examine the effect of propagation of posterior values and
uncertainties on the final emission estimates. We compare the base run to a run that has no propagation
(No_propagation and $CO_2$_only_no_propagation) but instead starts from the same prior mean and uncertainty on





each of our 14 days considered. The runs without would allow the parameter values to change over time. The
results are discussed in Sect. 3.3.
**Table 2. Overview of the inversions: which tracers are included, the length of the state vector and whether posterior**
**values and uncertainties are propagated.**

| Inversion name | Tracers | State vector length (per day) | Propagation to the next day |
|---|---|---|---|
| Base | All | 44 | Yes |
| Short_state | All | 21 | Yes |
| No_propagation | All | 44 | No |
| $CO_2$_only | $CO_2$ | 44 | Yes |
| $CO_2$_only_no_propagation | $CO_2$ | 44 | No |

**3 Results**
Before demonstrating the use of our dynamic emission model in an inverse framework, we demonstrate its
application as a simple but versatile method to generate hourly gridded emissions for multiple species with full
covariances.
**3.1 Dynamic emissions and their uncertainty**
The total yearly emission of $CO_2$ for the Netherlands calculated with the dynamic emission model is 180 Tg $CO_2$
with an uncertainty of 15 % (1-sigma Gaussian based on 500 members of a Monte Carlo simulation). This matches
the total of the Dutch national emission inventory for 2014 by design (step 1), but the uncertainty on the latter was
estimated with a similar Monte Carlo simulation to be only 1 % for $CO_2$ in 2004 (Ramírez et al., 2006). This
smaller uncertainty is fully due to the use of country-specific emission factors with a much smaller range than we
derived from the IEA and IPCC inventories. Spatial disaggregation (step 2) does not affect the uncertainty of the
domain aggregated annual fluxes, and the time profiles (step 3) have no impact on the yearly total emissions. For
CO, $NO_x$ and $SO_2$ the uncertainties in the dynamic emission model are much larger, with medians (CI's) of $6.5 \times 10^8$
($1.3 \times 10^8$–$6.8 \times 10^9$) kg CO yr$^{-1}$, $5.0 \times 10^8$ ($1.2 \times 10^8$–$5.1 \times 10^9$) kg $NO_x$ yr$^{-1}$, and $1.3 \times 10^8$ ($5.1 \times 10^6$–$2.2 \times 10^{10}$) kg $SO_2$ yr$^{-}$
$^1$. These ranges result from uncertainties in the assumed ratios of their release per unit of $CO_2$ emitted.
Below the annual scale, time profiles have an impact on the uncertainties as well. The daily emissions of the
Netherlands depend on the day and the season (Fig. 8) and range from 0.36 to 0.76 Tg $CO_2$ day$^{-1}$. The time series
shows a seasonal cycle with lower emissions during the summer. There is a clear weekly cycle with reduced
emissions during the weekend. The uncertainty in the total daily emission varies between 8 and 15 %, which is
similar to or lower than the uncertainty in the yearly total emissions. The explanation for these relatively low
uncertainties is that many uncertainties are temporally uncorrelated and their impacts on individual days partially
cancel out. Moreover, the largest sectors (coal-fired power plants and industry) already have a large uncertainty
and adding more uncertainty through the time profiles has little impact. Nevertheless, the uncertainties introduced
through the time profiles cause an uncertainty in daily $CO_2$ emissions of about 7 %, if the other uncertainties are
excluded from the analyses.



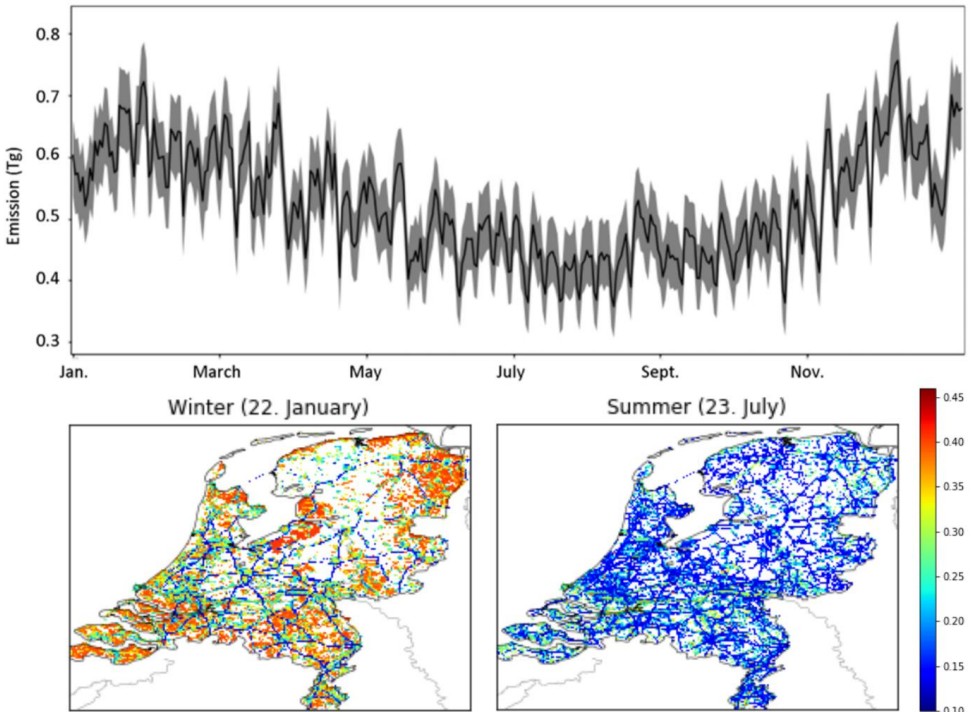

**Figure 8. (top) Time series of daily $CO_2$ emissions (in Tg $CO_2$ day$^{-1}$) and their uncertainty. Given is the interquartile range (shaded area) and the median (line) from the ensemble. (bottom) Map of annual mean relative uncertainty of emissions for the top 25 % pixels with the largest emissions, during a winter month (dominated by household gas- and electricity use) and a summer month (electricity and road-traffic dominated).**

Differences in the relative contribution of different sectors are evident when looking at the map of uncertainties across the Netherlands (Fig. 8), reflecting both the most uncertain parameters, but also the dominant source sectors. Winter emissions for example are dominated by household gas-usage, while industrial and traffic emissions give rise to uncertainty year-round at a 10–30 % level. We further identified the most important parameters per source sector with a Monte Carlo simulation per source sector (Fig. 9). Results shows that the road traffic and shipping sectors contain the smallest relative uncertainties, although the time profile for shipping causes an uncertainty of about 7 % in the total shipping emissions. The industrial emissions are most uncertain, and this is almost exclusively due to the emission factor, which causes an uncertainty of 41 % in the total industrial emissions. Similarly, the power plant emissions have a large relative uncertainty due to the uncertain emission factor of coal-fired power plants (19 %). Also, for households and glasshouses the emission factor is uncertain (14 % and 26 %, respectively), but here the time profiles also have a large impact (10 % and 16 %, respectively).



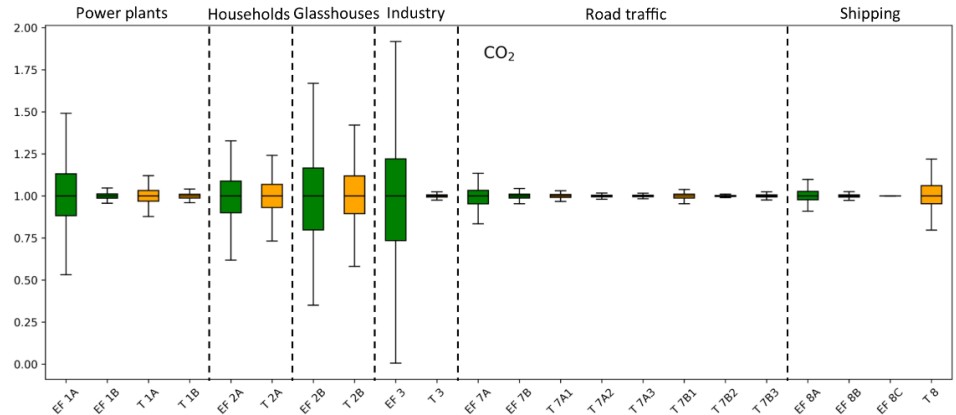

**Figure 9. Box plots showing the uncertainty in the $CO_2$ emissions from power plants (1A+1B), households (2A), glasshouses (2B), industry (3), road traffic (7A+7B) and shipping (8A+8B+8C) caused by individual parameters affecting that sector. Uncertainty is represented as the spread in daily (normalized) emissions from each ensemble member (N=500) over a full year (N=365). EF refers to an emission factor (green bars) and T to a time profile (orange bars). (Sub)sectors are indicated with their short names as summarized in Table 1. Note that the time profiles of road traffic emissions are specified per road type (1 = highway, 2 = main road, 3 = urban road). Minor parameters that have very small impacts on $CO_2$ emissions are not shown here (23 out of 44).**

## 3.2 Optimizing dynamic emissions

In the base inverse modelling setup, our system is able to improve the mean estimate and reduce the uncertainty on total $CO_2$, CO, $NO_x$, and $SO_2$ emissions. Figure 10 shows the probability density function of these estimated total emissions, compared to the prior (using parameters derived from IPCC/EEA) and the truth (created with country-specific parameter values). Interestingly, the posterior result deteriorates slightly when using a shortened state vector in which 11 parameters of "minor" influence (such as the $SO_2$:$CO_2$ ratio of household emissions) are not optimized from their incorrect prior. This is caused by sporadic atmospheric signals that are dominated by household emissions, even if these emissions only contribute a small fraction to the total emissions. These signals are then used to update the emission factor, while the emission ratios are also incorrect.





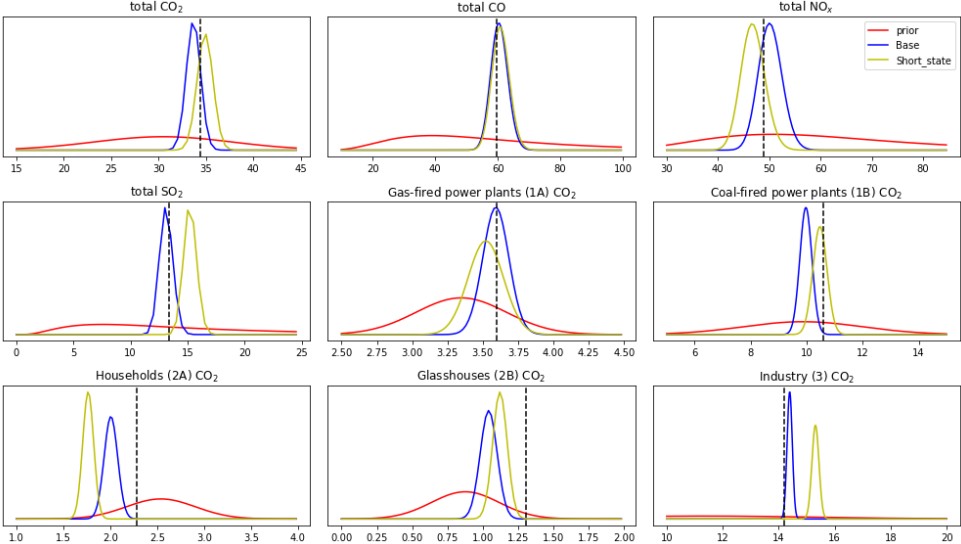

**Figure 10. Probability density functions of emissions per species or per source category (for CO₂) in units of Tg (CO₂) or Gg (CO, NOx, SO2). The truth is shown as a vertical dashed line, typically well-matched by the mean of the posterior in blue. Using a shortened state vector (yellow) deteriorates the total non-CO₂ emissions substantially and leads to misattribution of CO₂ emissions in minor categories such as 2A (households).**

With $CO_2$ as the only tracer in the inversion we find that we can still estimate total $CO_2$ emissions quite well (truth-minus-optimized = 0.03 Tg $CO_2$ yr$^{-1}$), but we lose the capacity to attribute emissions to specific sectors. Instead, mainly the emission factor of the largest single source being industry (EF3) is optimized. We illustrate this in Fig. 11, using the No_propagation run. The large spread across the 14 individual days indicates that the emission factor jumps around within a large prior uncertainty distribution and is not well-constrained on each day. Some of the other emission factors show almost no deviation from the prior and little variability. Given the constraints posed by $CO_2$ observations alone, and the limited number of parameters that change the simulated $CO_2$, optimizing EF3 improves the results at the lowest costs. Introducing the co-emitted species allows the system to identify the source of a residual, and attribute it to the right parameters if sufficient sensitivity is present. This is especially true for those sectors that have relatively small emissions and/or uncertainties, like 2B and 1A. This is corroborated by the posterior covariance matrices (See Appendix B) which show a reduction in parameter correlations for those parameters (i.e., a better mathematical separation of the estimates) when all tracers are included in the estimate. For other parameters the median values are further from the truth than the prior (e.g. for $R_{SO2}$ 8), which indicates that there is too little sensitivity to these parameters.





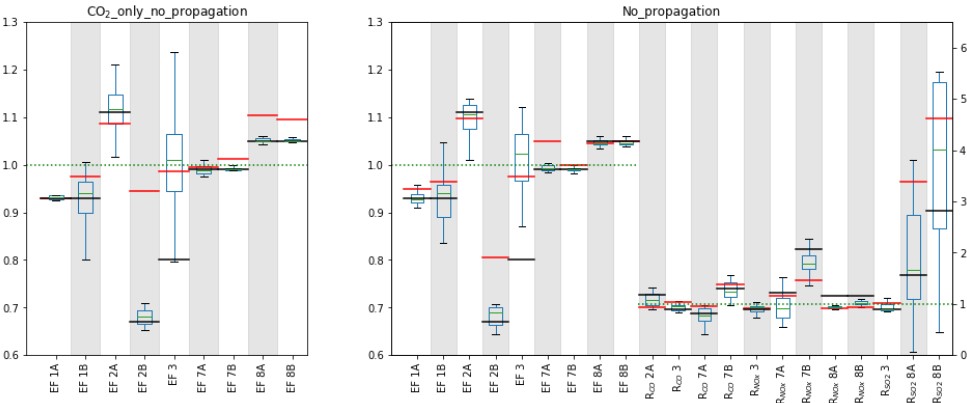

606

**Figure 11. Spread (Q1-Q3) and median values of the parameter scaling factors for the fourteen individual days included in the CO₂_only_no_propagation (left) and No_propagation (right) inversions, and final value of the CO₂_only (left) and base (right) inversion (red lines). The prior values are indicated by the black lines and the truth is indicated with the green dotted lines (value of 1.0). The left y-axis is for the emission factors, the right y-axis for the tracer ratios. The inversion with all tracers shows more variability in the emission factors and larger deviations from the prior values.**

### 3.3 Localization and propagation of information

Propagating information on parameter values from one day to the next is often better than using the median of individual days' estimates as illustrated by the red lines in Fig. 11. Nevertheless, the sporadic detection of plumes with specific signatures suggests that a form of selection or localization of the strongest signals could reduce noise and improve the estimate for the No_propagation run. We therefore ranked the 14 daily independent parameter estimates based on their relative posterior uncertainty and the residuals in an attempt to find the most trustworthy parameter values. This ranking is done per parameter, so the best estimate of different parameters can be related to different days. The increase in residual (same for all parameters) and posterior uncertainty (of the industrial emission factor) is shown in Fig. 12, where the 3–5 highest ranked days have similar characteristics after which the reliability decreases. On the lower ranked days, atmospheric signals from that particular source sector are too small (or even absent) to update the parameters related to that source sector. A similar pattern is found for the other parameters (not shown), with 2–5 days of high sensitivity out of 14.

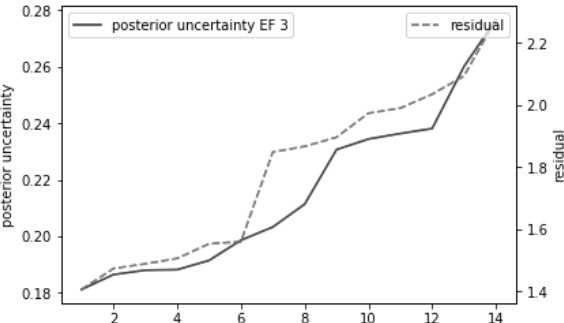


**Figure 12. Increase in posterior uncertainty (1σ of unitless scaling factor) in the industrial emission factor (EF 3) and absolute mean residual of CO₂ (in ppm) from highest- to lowest-ranked days.**






When we use the top-3 averaged parameter values to calculate emissions we find for most sectors that the emission
estimate is similar to the base run, albeit with a larger uncertainty, while for a few specific sectors results
deteriorate. This suggests that selecting for strong signals can dampen spurious noise, but still does not improve
on the base run that includes full propagation of the covariances, hence carrying information on parameter
correlations that is partially lost in the No_propagation run.
From the posterior covariance matrices we can confirm our selection of "good" days, as these typically show
relatively weak correlations between parameters. For the industrial sector (emission factor, $R_{NOx}$, $R_{SO2}$) these are
typically weak on most days, and indeed the mean over the entire period already gives a robust estimate of the true
parameter value (Fig. 13). The parameters with the strongest correlations are $R_{CO}$ of households and road traffic,
and their mean values tend to be dominated by a few outliers. Selecting days on which the posterior parameter
correlations are weak (i.e. the atmospheric signal clearly contains information about this specific parameter) results
in a large improvement compared to the prior or a 14-day average. Moreover, these results show a similar or better
performance as the top-3 selection based on Fig. 12 (0.08 for EF3 and 0.18 for $R_{co}$ 7A, not shown), and are closer
to the base run.

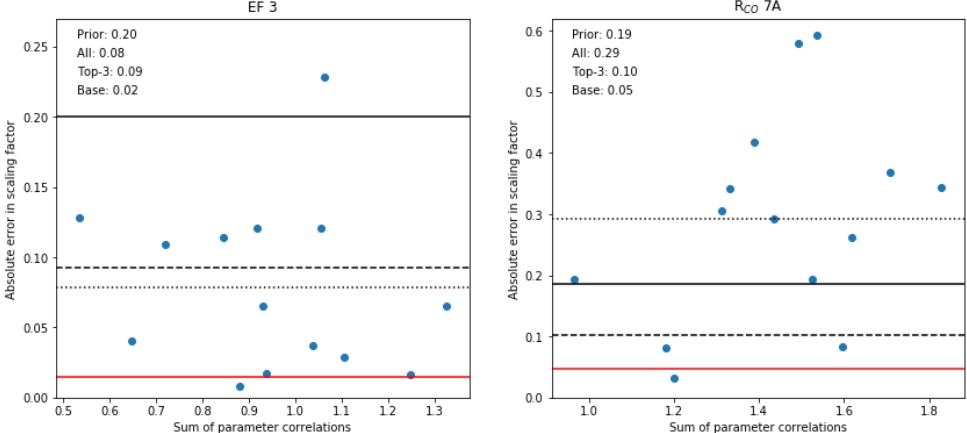


**Figure 13. Scatter plot of the absolute error in the scaling factor of the industrial emission factor (EF 3) and $R_{CO}$ of road**
**traffic (7A) against the sum of the parameter correlations of the same parameters. The correlation coefficients are -0.17**
**and 0.37 respectively. The horizontal lines give the average absolute error in the scaling factor for the prior (full black**
**line), if all 14 days are averaged (dotted line), and based on the 3 days with the smallest parameter correlations (dashed**
**line) and the result for the base run (full red line). The values are also given.**
**4 Discussion**
**4.1 Optimizing the dynamic emission model**
The dynamic emission model has the advantage over static emission fields that its parameters are optimized, giving
physical meaning to the results. To reduce the size of the problem, the state vector can be populated with those
parameters that are most important and/or uncertain. However, we find that uncertain, non-included parameters
can still significantly affect the optimization. Therefore, the size of the state vector should be considered carefully
when applying this method. Moreover, we performed an experiment to establish the possibility to optimize the
time profiles as part of the state vector. Although we found some improvements, it appears to be difficult to
differentiate between the different variables in Eq. (2) that have a linear relationship based purely on the



observations. Therefore, the results are not shown and optimizing the temporal dynamics of the emission model
requires further work.
Additionally, we identified the base run as the simplest method to get good estimates, but we do note that our
current propagation scheme does not yet include error growth. That means that eventually the ensemble will
converge on a parameter value and discard incoming observational evidence, unless the covariance is inflated to
allow new updates. Examples of such a covariance inflation scheme are ample in literature and in principle not
difficult to include, but were not yet considered in this work as the time periods covered were still short.
Finally, we have demonstrated that tracers are suitable for source attribution. Several previous studies have used
co-emitted species as tracer for fossil fuel $CO_2$ by taking advantage of the specific emission ratio characteristics
of each source sector (Lauvaux et al., 2013; Lindenmaier et al., 2014; Turnbull et al., 2015) and came to similar
conclusions. Nevertheless, the uncertainty in emission ratios remains a source of error and therefore the
optimization of emission ratios with our system is a promising step forward. Using co-emitted species to identify
the total fossil fuel contribution to the observed $CO_2$ signal is more difficult (Turnbull et al., 2006). The reason for
this is that there is a large variability in emission ratios between sectors. This makes it difficult to establish an
average emission ratio for an urban area, because it depends strongly on the relative contribution of each source
sector and may vary over time.
**4.2 Radiocarbon and background definition**
Therefore, a nice addition to this inversion system would be the inclusion of radiocarbon measurements. The
radiocarbon isotope ($^{14}CO_2$) can be used to simulate fossil fuel $CO_2$ records and has been applied successfully in
several inverse modelling studies (Turnbull et al., 2006; Levin and Karstens, 2007; Miller et al., 2012; Turnbull et
al., 2015; Basu et al., 2016; Wang et al., 2018). The radiocarbon measurements could be used directly in the
inversion (as we did with the co-emitted species) or be used to define a fossil fuel $CO_2$ record in advance (Fischer
et al., 2017; Graven et al., 2018). Our urban network detects average fossil fuel $CO_2$ signals of about 5 ppm with
peaks up to 50 ppm. This would result in $\Delta^{14}C$ signals (the ratio of $^{14}CO_2$ to $^{12}CO_2$) of around 13 up to 130 per
mille, which are certainly detectable with current techniques. However, observations of carbon isotopes are
expensive and currently not widely available, so their applicability is still limited. Besides $\Delta^{14}C$ other isotope
signatures and tracers can also provide additional information. For example, $^{13}CO_2$ and $O_2/N_2$ can give insight in
the dominant sources and sinks or fuel types (Lopez et al., 2013; Van der Laan et al., 2014) and as such be an
indicator for the transition from fossil fuels to biofuels. They might also help to separate between the stack
emissions of industry and coal- and gas-fired power plants.
An additional advantage of including the radiocarbon isotope is that the uncertainty in the background $CO_2$ can be
excluded, i.e. only the fossil fuel record is considered. Here, we choose to ignore the uncertainty in the background,
except in the definition of the covariance matrix $R$, and attribute all tracer residuals to the fossil fuel emissions.
Yet an incorrect definition of the background causes a large bias in the optimized emissions (Göckede et al., 2010).
There are also several other methods to deal with the non-fossil fuel related $CO_2$ signals. First, the uncertain
background can be added to the state vector and be optimized in the inversion. For example, He et al. (2018) have
shown that high-altitude aircraft observations are suitable to improve regional biosphere flux estimates by
correcting the bias in boundary conditions. Second, a mole fraction gradient over the area of interest can be
calculated using an upwind and downwind site such that the boundary inflow plays no role anymore (Turnbull et





al., 2015). This method was shown to reduce the impact of boundary inflow, but only when the wind direction is
more or less perpendicular to the gradient (Bréon et al., 2015; Staufer et al., 2016). Therefore, this method limits
the amount of useful measurements.

### 4.3 Error correlations

The dynamic emission model also allows us to study the correlations between model parameters, therefore giving
more insight in how information can be used in the system and which parameters are more challenging to separate.
Previously, Boschetti et al. (2018) have used the presence of error correlations between emissions of different
species and found that this reduces the posterior uncertainties for all species. They even show that the uncertainty
reduction increases with the correlation and that an incorrect definition of the error correlations may cause a
systematic bias in the posterior emission estimate. However, error correlations are only beneficial if the
atmospheric observations can distinguish between the correlated parameters. If this is not the case the presence of
parameter correlations can result in poorly constrained parameters and/or large posterior uncertainties. This is
especially true when parameters are sensitive to parameter correlations, as we show for $R_{CO}$ of road traffic.
An important question is then why some emission model parameters are more sensitive to the presence of
parameter correlations than others. One hypothesis is that parameters with a lower prior uncertainty are more
sensitive to the presence of parameter correlations. The idea behind this is that if we reduce the diagonal value
(uncertainty) by a factor of 4 the off-diagonal value (parameter correlation) reduces by a factor of 2. This means
that the parameter correlation is relatively stronger if the uncertainty is lower (Boschetti et al., 2018). This
hypothesis cannot be confirmed by our results, as we only find a correlation of -0.27 between the prior uncertainty
and the sensitivity to parameter correlations (defined as the correlation between the posterior uncertainty and the
sum of the parameter correlations). The main difficulty here is that not all parameters can be discerned with the
observed atmospheric signals. Although we included the additional co-emitted tracers for source attribution, the
emission ratios have a large uncertainty and the system can have difficulties assigning residuals to either the
emission ratio or the emission factor. Yet if we calculate an average sensitivity and total posterior uncertainty per
sector (by combining the emission factor and emission ratios per sector) we find a correlation coefficient of -0.82.
This suggests that this hypothesis might indeed be correct and source sectors with larger parameter uncertainties
are less sensitive to the presence of parameter correlations.

### 4.4 Atmospheric transport model errors

In addition to the experiments described in Sect. 2.3 we conducted an experiment that focused on the role of
transport model errors by using observed meteorology to drive the OPS model in the inversion. Like many authors
before us (McKain et al., 2012; Brioude et al., 2013; Lauvaux et al., 2013; Bréon et al., 2015; Boon et al., 2016)
we found a large impact on the performance of our system and once again confirmed the need for accurate transport
models. This experiment is not further shown in this work because of its redundancy with previous conclusions.
Nevertheless, we performed this experiment to examine whether transport errors are important when the state
vector consists of parameters that are valid for the entire domain. Random errors, such as errors in the wind
direction, are unlikely to affect the optimized emissions much when averaged over a longer time period and
domain. This was shown by Deng et al. (2017), who found little variation in the average $CO_2$ emission for
Indianapolis using different configurations of WRF to calculate the transport. However, they did find an impact





on the spatial distribution of the emissions. This becomes important when optimizing a specific source sector that
is clustered in one place, such as the glasshouses. We found that the glasshouse sector is only correctly optimized
with a specific wind direction. If the modelled wind direction is wrong the residuals would thus not be attributed
to the glasshouse sector as it is not in the modelled footprint of the measurement site. As such, we conclude that
the footprint definition has an impact on the optimized parameters, despite that the parameters have no spatial
distribution. Similarly, Broquet et al. (2018) mention that the location and structure of a simulated urban plume
might differ significantly from the true plume characteristics due to errors in the simulated wind speed and wind
direction.
Systematic errors, whether in the modelled transport or in the observations, are more difficult to solve as they do
not cancel out when simulating a longer period, and this can lead to biased emission estimates (Meirink et al.,
2008; Su et al., 2011). Several methods have been suggested to overcome problems with an incorrect description
of atmospheric transport, such as using an ensemble of atmospheric transport model simulations (Angevine et al.,
2014) or the assimilation of meteorological observations (Lauvaux et al., 2013). The latter showed lower biases in
buoyancy and mean horizontal wind speed. Another method that is often used is the selection of well-mixed
afternoon hours to exclude stable conditions under which pollutant dispersion is often poorly represented (Lauvaux
et al., 2013; Bréon et al., 2015; Boon et al., 2016). Such data selection however leads to a bias in the estimated
emissions when the diurnal cycle is not correctly accounted for (Super et al., 2019).
Here, we also applied a daytime selection criterion to mimic this situation. However, we found that night time
hours could be very useful to constrain our emissions. In our DA system we use residual fossil fuel enhancements
over a background (prior - true mole fraction enhancement) to constrain the fossil fuel fluxes. The larger the
residual, the more information can be gained from it since the impact of the observation error ($R$ matrix) is
relatively small. If, for example, the industrial emission factor is underestimated by 10 %, the residual industrial
enhancement (given a linear relationship between the emission factor and the total emission from this sector) will
be 10 % of the pseudo-observed mole fraction. This means that a large signal from the industry is needed to reach
a residual that is larger than the observation error ($\sigma$ is 1.6 ppm for $CO_2$). Looking at the time series of pseudo-
observations we find that such large signals mostly occur during night time or in the early morning. Therefore, the
inversion could benefit strongly from an improved description of night time boundary layers and stable conditions,
so that the large night time enhancements can be used to constrain the fossil fuel fluxes.
**5 Conclusions**
The aim of this study was to examine how well our DA system can quantify urban $CO_2$ emissions per source
sector. Since the prior consists of a dynamic fossil fuel emission model the model parameters are optimized rather
than the emissions themselves. The parameters are related to specific source sectors and to attribute residuals to
these sectors measurements of additional tracers (CO, $NO_x$ and $SO_2$) are included in the inversions. We tested this
system to examine its ability to overcome some major limitations in current urban-scale inversions: source
attribution, definition of the prior and its uncertainties, and the sensitivity to errors in atmospheric transport.
We find that inverse modelling at the urban scale is feasible when the observations contain a lot of information
about the different source sectors. When only $CO_2$ mole fractions are used in the inversion the total $CO_2$ emission
are well-constrained. Additional tracers are an important addition to the inversion framework in order to discern
the information belonging to specific source sectors and emission model parameters. However, even more tracers



might be needed to fully capture the heterogeneity of the emission landscape. Moreover, we argue that a dynamic emission model has some major advantages over regular emission maps, allowing us to constrain physically relevant parameters even in the absence of good prior information.

Nevertheless, quite some challenges remain. Transport modelling at this small scale needs to be improved to be able to use real urban observations, as under current conditions the transport error strongly dominates the results. Especially improving the description of night time boundary layers could be beneficial, because large atmospheric signals mostly occur during the period. For the future, additional advances need to be made to include satellite observations in the inverse modelling framework. The advantage of satellite data is that it covers data-sparse regions and with a larger view it can differentiate between the urban dome with high pollution levels and the cleaner rural areas, which is a nice addition to in situ measurements.

**Code and data availability**

The availability of the CTDAS (v1.0) code is described in a previous publication (Van der Laan-Luijkx et al., 2017), which forms the basis of the system described in this paper. Minor changes have been made to include the dynamic emission model. Revised code and the additional module used to describe the dynamic emission model and the creation of pseudo-observations is included as Supplement, as is a script used for the emission uncertainty analysis (Monte Carlo simulation). Input data for the dynamic emission model are taken from open, online databases and are summarized in Appendix A, including their data sources. Example input files for CTDAS and the OPS model are also included as Supplement.





**Appendix A**

**Table A1. Overview of all parameters in the dynamic emission model, their unit, function type, expected value and uncertainty (range).**

| Parameter | (Sub)sector | Unit | Function type | Expected value | Uncertainty |
|---|---|---|---|---|---|
| **Emission factor[a]** | Coal-fired power plants[c] | kg PJ$^{-1}$ | Normal | 1.01E8 | 23 % |
| | Gas-fired power plants[c] | kg PJ$^{-1}$ | normal | 5.61E7 | 10 % |
| | Households[c] | kg PJ$^{-1}$ | normal | 5.89E7 | 14 % |
| | Glasshouses[c] | kg PJ$^{-1}$ | normal | 5.61E7 | 25 % |
| | Industry[d] | kg PJ$^{-1}$ | normal | 7.66E7 | 40 % |
| | Road traffic cars[e] | kg PJ$^{-1}$ | normal | 7.24E7 | 10 % |
| | Road traffic HDV[e] | kg PJ$^{-1}$ | normal | 7.33E7 | 5 % |
| | Ocean shipping[f] | kg PJ$^{-1}$ | normal | 7.76E7 | 5 % |
| | Inland shipping[f] | kg PJ$^{-1}$ | normal | 7.30E7 | 5 % |
| | Recreational shipping[f] | kg PJ$^{-1}$ | normal | 7.10E7 | 5 % |
| **Emission ratio CO:CO$_2$** | Coal-fired power plants[e] | kg kg$^{-1}$ | lognormal | 1.29E-4 | 8.7E-7–2.9E-4 |
| | Gas-fired power plants[e] | kg kg$^{-1}$ | lognormal | 8.47E-4 | 3.4E-4–2.5E-3 |
| | Households[e] | kg kg$^{-1}$ | lognormal | 3.88E-3 | 8.3E-4–9.6E-3 |
| | Glasshouses[e] | kg kg$^{-1}$ | lognormal | 5.40E-4 | 3.1E-5–7.7E-4 |
| | Industry[d] | kg kg$^{-1}$ | normal | 2.06E-3 | 40 % |
| | Road traffic cars[e] | kg kg$^{-1}$ | lognormal | 1.32E-2 | 8.0E-5–6.5E-2 |
| | Road traffic HDV[e] | kg kg$^{-1}$ | lognormal | 2.22E-3 | 9.3E-5–1.3E-2 |
| | Ocean shipping[f] | kg kg$^{-1}$ | normal | 2.32E-3 | 30 % |
| | Inland shipping[f] | kg kg$^{-1}$ | normal | 3.42E-3 | 30 % |
| | Recreational shipping[f] | kg kg$^{-1}$ | normal | 2.96E-1 | 30 % |
| **Emission ratio NO$_x$:CO$_2$** | Coal-fired power plants[e] | kg kg$^{-1}$ | lognormal | 5.94E-4 | 3.0E-4–9.4E-4 |
| | Gas-fired power plants[e] | kg kg$^{-1}$ | lognormal | 2.00E-3 | 2.6E-4–3.7E-3 |
| | Households[e] | kg kg$^{-1}$ | lognormal | 1.50E-3 | 4.8E-4–3.3E-3 |
| | Glasshouses[e] | kg kg$^{-1}$ | lognormal | 1.63E-3 | 5.0E-4–3.5E-3 |
| | Industry[d] | kg kg$^{-1}$ | normal | 6.56E-4 | 40 % |
| | Road traffic cars[e] | kg kg$^{-1}$ | lognormal | 1.76E-3 | 9.0E-5–7.5E-3 |
| | Road traffic HDV[e] | kg kg$^{-1}$ | lognormal | 1.11E-2 | 3.3E-4–3.7E-2 |





| | | | | | |
|---|---|---|---|---|---|
| | Ocean shipping[f] | kg kg⁻¹ | normal | 2.32E-2 | 30 % |
| | Inland shipping[f] | kg kg⁻¹ | normal | 1.37E-2 | 30 % |
| | Recreational shipping[f] | kg kg⁻¹ | normal | 1.97E-3 | 30 % |
| **Emission ratio SO₂:CO₂** | Coal-fired power plants[e] | kg kg⁻¹ | lognormal | 1.66E-4 | 2.9E-5–4.4E-4 |
| | Gas-fired power plants[e] | kg kg⁻¹ | lognormal | 5.01E-6 | 2.9E-6–7.2E-6 |
| | Households[e] | kg kg⁻¹ | lognormal | 2.21E-5 | 1.4E-5–6.7E-5 |
| | Glasshouses[e] | kg kg⁻¹ | lognormal | 8.91E-6 | 5.2E-6–1.3E-5 |
| | Industry[d] | kg kg⁻¹ | normal | 4.28E-4 | 40 % |
| | Road traffic cars[g] | kg kg⁻¹ | normal | 1.01E-6 | 100 % |
| | Road traffic HDV[g] | kg kg⁻¹ | normal | 8.16E-7 | 100 % |
| | Ocean shipping[f] | kg kg⁻¹ | lognormal | 6.18E-3 | 3.3E-4–2.0E-2 |
| | Inland shipping[f] | kg kg⁻¹ | lognormal | 6.57E-3 | 3.5E-4–3.0E-2 |
| | Recreational shipping[f] | kg kg⁻¹ | lognormal | 3.14E-4 | 1.1E-4–7.0E-4 |
| **Hourly time factor[h]** | Coal-fired power plants | - | normal | 1 | 28 % |
| | Gas-fired power plants | - | normal | 1 | 43 % |
| | Industry | - | normal | 1 | 5 % |
| | Households | - | normal | 1 | 43 % |
| | Glasshouses | - | normal | 1 | 74 % |
| | Road traffic cars highway | - | normal | 1 | 18 % |
| | Road traffic cars main road | - | normal | 1 | 18 % |
| | Road traffic cars urban road | - | normal | 1 | 18 % |
| | Road traffic HDV highway | - | normal | 1 | 41 % |
| | Road traffic HDV main road | - | normal | 1 | 18 % |
| | Road traffic HDV urban road | - | normal | 1 | 48 % |
| | Total shipping | - | normal | 1 | 31 % |
| **Energy consumption per activity data[i]** | Total power plants | PJ/mln € | - | 8.22E-4 | - |
| | Households | PJ/dd[b] | - | 0.199 | - |
| | Glasshouses | PJ/dd[b] | - | 0.061 | - |
| | Industry | PJ/mln € | - | 7.05E-4 | - |
| | Road traffic cars | PJ/mln € | - | 3.98E-4 | - |





| | Road traffic HDV | PJ/mln € | - | 2.01E-4 | - |
|---|---|---|---|---|---|
| | Total shipping | PJ/mln € | - | 1.51E-4 | - |
| **Fraction of total energy consumption per subsector[j]** | Total power plants: coal | - | - | 0.62 | - |
| | Total power plants: gas | - | - | 0.38 | - |
| | Road traffic cars: highway | - | - | 0.47 | - |
| | Road traffic cars: main road | - | - | 0.28 | - |
| | Road traffic cars: urban road | - | - | 0.25 | - |
| | Road traffic HDV: highway | - | - | 0.56 | - |
| | Road traffic HDV: main road | - | - | 0.24 | - |
| | Road traffic HDV: urban road | - | - | 0.20 | - |
| | Total shipping: ocean | - | - | 0.79 | - |
| | Total shipping: inland | - | - | 0.20 | - |
| | Total shipping: recreational | - | - | 0.01 | - |

[a] Emission factor for coal-fired and gas-fired power plants include uncertainty due to variations in fuel type, including burning
of biomass (5 % uncertainty). For households assume 8 % wood combustion based on $CO_2$ emission values (*Vernieuwd*
*emissiemodel houtkachels*, by B.I. Jansen (TNO, 2016)), the remainder is natural gas (with 10 % uncertainty). For glasshouses
assume only natural gas combustion, including 20 % additional uncertainty due to use of cogeneration plants. For road traffic
cars assume 69 % gasoline, 29 % diesel and 2 % LPG (with 5 % uncertainty); for road traffic HDV assume 100 % diesel.
[b] dd = degree day
[c] Expected value and uncertainty based on IPCC Emission Factor Database (EFDB) using 2006 IPCC guidelines
[d] Expected value based on Emissieregistratie (emission) and CBS (energy consumption); uncertainty based on expert
judgement
[e] Expected value and uncertainty based on the EMEP/EEA air pollutant emission inventory guidebook 2016
[f] Expected value and uncertainty based on *$CO_2$, $CH_4$, and $N_2O$ emissions from transportation-water-borne navigation*, by Paul
Jun, Michael Gillenwater, and Wiley Barbour (Good Practice Guidance and Uncertainty Management in National Greenhouse
Gas Inventories)
[g] Expected value based on Air Pollutant Emission Factor Library (Finish Environment Institute); uncertainty based on expert
judgement
[h] Uncertainties based on comparison activity data-based time profiles and estimated time profiles from environmental/socio-
economic factors
[i] Expected value based on CBS (energy consumption, GDP) and KNMI (degree day sum)
[j] Expected value based on Emissieregistratie



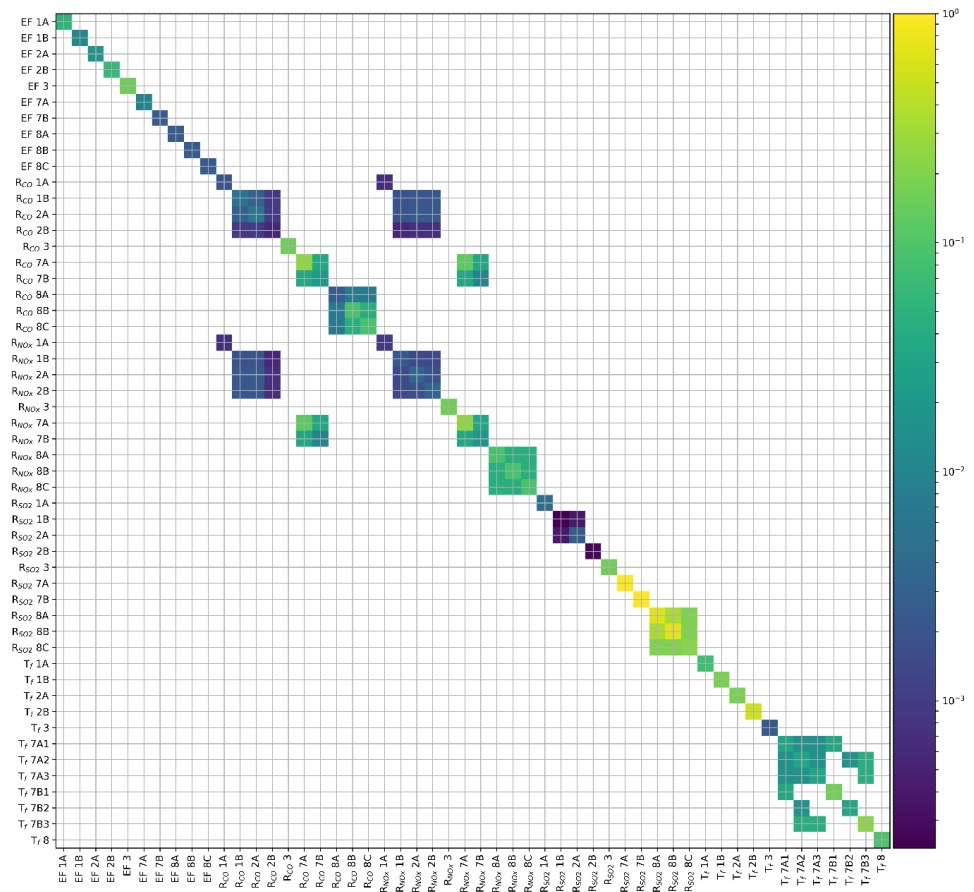


**Figure A1. Covariance matrix for all parameters in the dynamic emission model. For all covariances we assume a correlation coefficient of 0.5. (Sub)sectors are indicated with their short names as summarized in Table 1. Note that the time profiles of road traffic emissions are specified per road type (1 = highway, 2 = main road, 3 = urban road).**



**Appendix B**

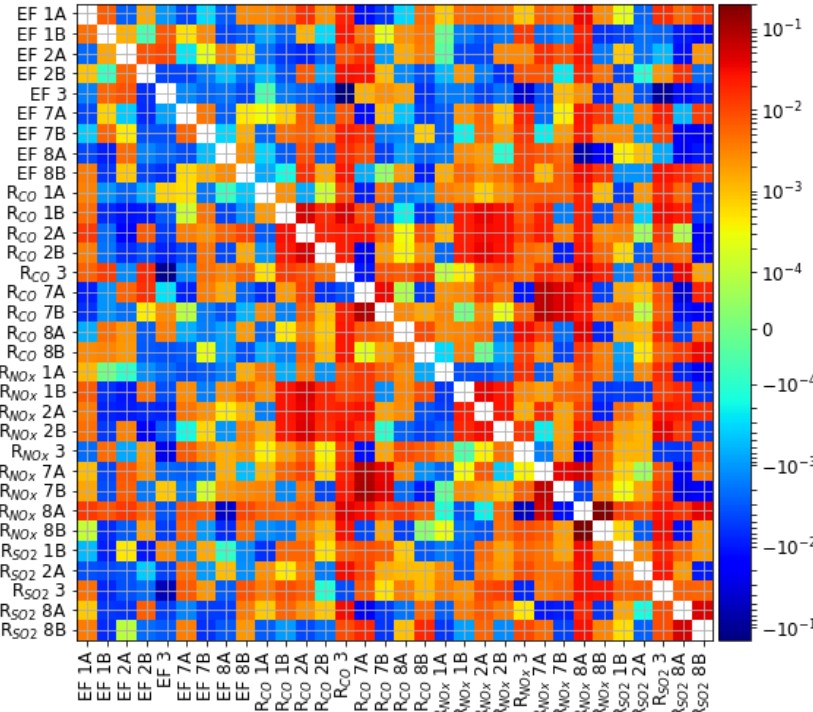

**Figure B1. Matrix showing the difference in correlation coefficient (r) between the $CO_2$_only_no_propagation and No_propagation run averaged for all 14 days, where positive differences indicate reduced parameter correlations when all tracers are included (No_propagation). (Sub)sectors are indicated with their short names as summarized in Table 1. For some parameters a strong reduction in parameter correlations is shown, indicating that with all tracers that parameter can be more easily separated from others, for example the emission factors of industry and coal-fired power plants (EF3 and EF1B).**

**Author contribution**

The initial ideas are developed by WP, IS, HACDvdG and MKvdM. IS and SNCD developed the dynamic

emission model. IS and WP are responsible for setting up the inverse modelling experiments and prepared the

manuscript with contributions from all co-authors.

**Competing interests**

The authors declare that they have no conflict of interest.

**Acknowledgments**

This study was supported by the VERIFY project, funded by the European Union's Horizon 2020 research and

innovation programme under grant agreement No 776810; partly funded by EIT Climate-KIC project Carbocount-





CITY (APIN0029 2015-3.1-029 P040-04) and the EIT Climate-KIC Fellows programme (ARED0004 2013-1.1-
008 P017-0x).

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
