# Peer review of "Optimizing a dynamic fossil fuel CO2 emission model with 1"

_Geoscientific Model Development, 2019_

## Referee Comment (RC1) · Anonymous Referee #1 · 16 Dec 2019

This manuscript presents a modelling framework to optimize fossil fuel CO2 emissions using a data assimilation system and atmospheric observations. The prior emissions are estimated using a dynamic CO2 emission model, which allows constraining physically relevant parameters. The manuscript provides a novel approach, that can overcome some current limitations in urban-scale inversions such as source attribution, definition of the prior emissions and its uncertainties, and the sensitivity to errors in atmospheric transport.

The paper is well written and clear and a very good contribution for GMD. Results are presented in a detailed way and the conclusions are well-reasoned. My only major comment has to do with some of the subsections of the methods sections, which sometimes are not presented in sufficient detail and/or remain a little bit too general.

1. Section 2.1.1 and Table A1: How is the "E/A" term derived from the IEA statistics (L175)? According to Table A1, "E/A" values are derived from CBS and KNMI (description of these acronyms should be provided). To the best of my knowledge, the information that IEA reports is primary energy consumption by sector and fuel, which is equivalent to the "E" term of equation 1. Should not it be more efficient to directly use the "E" information provided by IEA instead of deriving it from the expression A * (E/A) proposed in equation 1? I find difficult to understand what is the added value of having to compile the "A" and "E/A" terms instead of directly using "E". Also, when describing "A" some examples are used such as "vehicle kilometers driven" (L161), but according to Table A1, the units used for the term "E/A" in road traffic cars and HDV are "PJ/mln€'. Should not it be "PJ/km"? The "A" terms and corresponding sources of information should also be provided in Table A1.

2. Section 2.1.3: This section remains too general, especially when compared to the previous one, where the temporal disaggregation methodology is presented in a detailed way for each sector. It is not clear to the reader the specific datasets/methods that are being used to spatially distribute the emissions for each sector. More details should be provided (perhaps the spatial proxies used should also be summarized in Table A1). Later on, in the manuscript, the authors say that the spatial distribution is assumed to be well-known (L346) and therefore this element is not considered when performing the uncertainty analysis. This sentence however seems at odds with a previous statement, which says that "their uncertainty increases rapidly when disaggregating them towards finer spatiotemporal resolutions" (L52). Considering the special increase in the emissions uncertainty that the introduction of spatial disaggregation generally causes, the non-inclusion of this element in the uncertainty analysis should be better justified (i.e. better discussed why the spatial proxies applied in this study can be assumed to be well-known).

In addition to these major comments, I list several doubts and minor comments mostly related to suggestions to improve the presentation of the work:

L94: Change (Andres et al., 2016) (Super et al., 2019) for (Andres et al., 2016; Super et al., 2019)

L104: I think that the concept of "near real-time" is too strong. For instance, this would imply that traffic emissions are estimated based on near-real time data collected from traffic counts and, therefore, that congestion situations or traffic accidents are considered when calculating the dynamic emissions. A similar thing would apply to power plants (e.g. emissions are derived from near-real time collected data on the activity of each individual facility).

L120: Replace "inverse part" for "inverse modelling part"

Table 1: Could you provide a reference to the CO2 contribution shares that are shown in Table 1?

L206: Some European studies have suggested the use of 15.5°C as the value for defining the threshold temperature when calculating the HDD (e.g. https://rmets.onlinelibrary.wiley.com/doi/epdf/10.1002/joc.3959). According to the results shown in Figure 3 (left), the parametrization proposed for households (18°C) is underestimating most of the observed peaks in winter, while it overestimates the ones observed during spring/summer. On the contrary, the parametrization proposed for glasshouse (15°C) reproduces much better the winter peaks. Do you think that reducing the value of Tb for the household parametrization could allow improving the reproduction of winter peaks? (this is just a suggestion, does not need to be added in the revised manuscript)

L220: Are you referring to the MACC-III fixed temporal profile? Please specify

L239: Add a reference to the ENTSO-E database (e.g. https://www.sciencedirect.com/science/article/pii/S0306261918306068)

L244: The correlations presented between power generation and meteorological variables are rather low. This implies that the proposed parametrization for this sector is not so well correlated with observed activity data such as it is for other sectors (e.g. households or road transport). Considering the importance of this sector to the total $CO_2$ emissions, perhaps it would be interesting to discuss how these parametrizations could be improved in future works.

L260: Could you also provide the $R^2$ value for daily data?

Table 1 / L272 / Figure 9: The industrial sector is the largest contributor to total $CO_2$, but at the same time is the only sector that has not been split between subcategories. Is there a specific reason for that? Should not a split between e.g. type of industries would help to provide better temporal parametrizations or reduce the uncertainty of the emission factors for this sector?

Figure 5 (left): It looks like the activity data (red line) is missing for the last day

Figure 9: According to this figure, the uncertainty of the time profile "T" is larger in the household sector than in the power plant sector. Nevertheless, the correlation between the temporal parametrization and true activity data reported for the household sector is higher than the one reported for the power plant sector. Is there a specific explanation for that?

Section 3.1: I assumed that the meteorological-dependent time profiles were calculated using the WRF model, but perhaps it should be clarified at some point in this section.

Section 5: In the introduction section the authors pose three research questions that want to answer with this study. It would be interesting to rewrite the conclusions section so that it provide concise and clear statements that directly answers each one of these research questions (i.e. include a bullet list with a statement per question). This structure may facilitate the reader to link the posed questions with the outcomes of the

work.

---

## Referee Comment (RC2) · Anonymous Referee #2 · 17 Feb 2020

The paper describes a new modelling framework to describe urban fossil fuel emissions of CO2 (ffCO2) in which emission ratios vary in time in space. To achieve this, the authors use atmospheric gases that are co-emitted with ffCO2 and range of proxy data that are associated with typical sectors that lie within the urban domain. They apply the resulting framework to a synthetic numerical experiment focused on the Rijnmond area, Netherlands.

This is a nice piece of work that with some development will eventually address some of the outstanding challenges we face as a community to quantify urban ffCO2. My recommendation is to accept the manuscript for publication after the authors have ad-

dressed my comments.

Broad comments

This is a chunky piece of work that contains a lot of information. For the sake of readability I encourage the authors to consider judicious use of additional appendices.

I have seen the authors present this work before and the use of "dynamic" has always rankled me. They could have just as easily described their new inventory as an online model that is fed with time-dependent data with resulting emissions being passed directly to subsequent atmospheric calculations. This is in contrast with static or offline inventories. Static inventories can also be dynamic in time and space, albeit on a discrete basis.

The figures are of low quality. Not sure why. I could barely read the text in Figure 1 and many of the other figures are grainy. Better quality figures will ultimately make the work easier to appreciate.

Figures would also benefit from being labeled directly, e.g A), B), C), etc. In some instances when columns are rows show something common a well-placed label would be useful. For example, Figure 4 would benefit from "Gas fired" and "Coal fired" labels for the top left and top right labels.

Bug bear: kindly please refrain from using "quite" as a descriptor throughout your paper. It is scientifically meaningless. Focus on the statistics that often accompany your statements.

Specific comments

Line 141: reason for greenhouses would be welcome here. Please mention tomatoes later but introduce the usage here.

Section 2.2.1. I think there might be a problem with units in your equation. Flux Fx should be mass/time but units of the contributing variables don't result in that unit.

Please clarify units for all variables shown in equation 1.

Figure 2. Please make this bigger.

Lines 203- 216 describe the definition of the time factor. I found the exposition of this point opaque, especially the accompanying mathematics. Please expound your argument.

Figure 3. The drop in relative gas consumption during May-Sept presumably reflects warmer weather. Are the spikes during this period due to cold days?

Pages 8-9 I was unclear reading through this text how much was based fact, e.g. the reason behind gas-fired power plants (weakly) negatively correlated with wind speed, and how much was interpretation. Please clarify. Generally, this reader would appreciate a summary table that explains which variables are being used as proxy data for various urban sector emissions.

Curiosity: are gas-fired power plants quicker to respond to shortfalls in energy provision than coal-fired plants? Does this explain the weaker correlation reported in lines 259-261?

Uncertainty analysis shown in Section 2.4.1. is important for inverse modellers. Is this a stop-gap approach or do you envisage this as a final method?

Section 2.2. Convention dictates that vectors and matrices are denoted as emboldened lower- and upper-case variables, respectively.

Section 2.2.1. There is a lot being described here. Worth a schematic?

Section 2.2.3. Closed-loop numerical experiments are considered useful only if the truth and prior are independent in some way. Some calculations might use independent inventories while others use independent transport models. Using the "dynamic" version of the static inventory is not sufficiently independent (e.g. Figures 5 and 6). Consequently, the authors have presented a very optimistic scenario. At least, the

author should acknowledge this situation.

Section 2.2.3. The authors assume no contribution from biogenic CO2 to the excess CO2 over the background. This is not a general assumption. How will they cope with an urban area with parks, for example?

Section 2.2.3. A few more details are necessary to describe the data. Ideally, earlier in the manuscript. I am surprised that the authors can achieve what they have with a handful of data collected at 10 metres a.s.l. Maybe this can only work in the Netherlands? Also, what is the origin of the values used in the R matrix?

Section 3. State that CI = confidence interval. Also, clarify "Below the annual scale" on line 543.

Section 3.2. The result associated with a shortened state vector was interesting and something this reviewer had not considered fully. How do we decide on the correct length of the state vector? Will this be location specific?

Minor comment: avoid using yellow in figures (Figure 10).

Figure 11 would benefit from a legend. It contains a lot of information that was all in the text and figure panel but it took a while to pick through it all.

Line 650. I would say that this approach provides a more detailed physical meaning of the results compared to estimating emission estimates.

Line 652. Non-included parameters?

Line 659. If your online inventory is using weather data to drive variations then you could use the correlation lengths associated with weather systems?

Section 4.2. Putting all your eggs in one basket with radiocarbon is not a wise move. It is one weapon in your arsenal. With the growth of biofuel combustion in urban regions, there will be a lot of combustion CO2 that is missed using radiocarbon. Something to consider in your discussion, especially since your group has just published work on this

topic that makes my point.

Line 774. Are you saying that your model has an advantage because it uses a source of information (emission-related parameters) that is often neglected by emission inventories?

---

## Author Comment (AC1) · 12 Mar 2020

We would like to thank the reviewers for their interest in our study and for expressing their thoughts on our work. We are aware of the large amount of work that we describe and the review comments have been helpful in reflecting on our work and pointing out parts that required more explanation. Below we address specific issues mentioned by the reviewers point by point. The manuscript has been updated accordingly (changes are highlighted, line numbers refer to the final manuscript).

*Anonymous Referee #1

*This manuscript presents a modelling framework to optimize fossil fuel CO2 emissions using a data assimilation system and atmospheric observations. The prior emissions are estimated using a dynamic CO2 emission model, which allows constraining physically relevant parameters. The manuscript provides a novel approach, that can overcome some current limitations in urban-scale inversions such as source attribution, definition of the prior emissions and its uncertainties, and the sensitivity to errors in atmospheric transport.*

*The paper is well written and clear and a very good contribution for GMD. Results are presented in a detailed way and the conclusions are well-reasoned. My only major comment has to do with some of the subsections of the methods sections, which sometimes are not presented in sufficient detail and/or remain a little bit too general.*

*1. Section 2.1.1 and Table A1: How is the "E/A" term derived from the IEA statistics (L175)? According to Table A1, "E/A" values are derived from CBS and KNMI (description of these acronyms should be provided). To the best of my knowledge, the information that IEA reports is primary energy consumption by sector and fuel, which is equivalent to the "E" term of equation 1. Should not it be more efficient to directly use the "E" information provided by IEA instead of deriving it from the expression A\* (E/A) proposed in equation 1? I find difficult to understand what is the added value of having to compile the "A" and "E/A" terms instead of directly using "E". Also, when describing "A" some examples are used such as "vehicle kilometers driven" (L161), but according to Table A1, the units used for the term "E/A" in road traffic cars and HDV are "PJ/mln C´". Should not it be "PJ/km"? The "A" terms and corresponding sources of information should also be provided in Table A1.*

We thank the reviewer for pointing out this ambiguity. The equation was adopted from Raupach et al. (2007) because of its simplicity and global applicability. The authors of that paper used this equation to calculate the total emission per country. Here, we apply the equation to each of the source sectors. Indeed, the energy consumption data could be used directly when such data are available. However, this is not always the case, while the term 'A', often GDP, is known for each country and we choose to use the full equation to ensure global applicability. For the same reason we suggested to use IEA statistics to make an assumption on the value of 'E/A' in absence of country-specific energy consumption data, for example by taking values from a comparable region. If country-specific data is available with more detail than IEA data, this can also be used of course, which is what we did here.

We have clarified why we used 'E/A' instead of 'E' in lines 183-184 and the suggestion to use IEA statistics (or other data) to estimate 'E/A' in lines 186-188. We replaced the acronyms with full names in lines 822 and 832-833. To avoid confusion, the example of vehicle kilometers driven is moved to lines 184-185, where it is

mentioned as an option to improve local emission estimates. The sources of 'A' are given in the footnote of Table A1, where it is mentioned that GDP is provided by Statistics Netherlands and degree day sum is based on data from the Royal Netherlands Meteorological Institute. For completeness, their values are added.

*2. Section 2.1.3: This section remains too general, especially when compared to the previous one, where the temporal disaggregation methodology is presented in a detailed way for each sector. It is not clear to the reader the specific datasets/methods that are being used to spatially distribute the emissions for each sector. More details should be provided (perhaps the spatial proxies used should also be summarized in Table A1). Later on, in the manuscript, the authors say that the spatial distribution is assumed to be well-known (L346) and therefore this element is not considered when performing the uncertainty analysis. This sentence however seems at odds with a previous statement, which says that "their uncertainty increases rapidly when disaggregating them towards finer spatiotemporal resolutions" (L52). Considering the special increase in the emissions uncertainty that the introduction of spatial disaggregation generally causes, the non-inclusion of this element in the uncertainty analysis should be better justified (i.e. better discussed why the spatial proxies applied in this study can be assumed to be well-known).*

We have carefully evaluated the reviewer's comment and distilled two main questions.

First, it seems unclear how the spatial disaggregation was performed and which data were used. We did not intend to improve the spatial distribution used in local inventories and decided to take over existing spatial patterns instead of creating our own. The reason is that spatial downscaling has received a lot of attention from inventory builders and instead we wanted to put our effort in improving the temporal downscaling as a first step towards building a dynamic emission model. For the Netherlands an emission map is available at 1x1 km$^2$ resolution and for many European countries such maps are exist. These maps are often based on widely available proxies for spatial disaggregation, like population density and land use type, although in some cases scaling factors are applied based on local circumstances. In absence of local knowledge, these proxies can be used directly and that is why we listed those in Section 2.1.3 to describe a methodology that is applicable in other regions as well.

Second, the reviewer raises the concern that the uncertainty in spatial disaggregation is not taken into account, while it is considered an important contributor to the overall uncertainties in a high-resolution emission map. We agree with the reviewer that these statements seem to be contradictory and that spatial downscaling indeed increases the uncertainty drastically. Our choice to not include the spatial component in our setup is further justified by the choice of observation network here, where we only consider 7 sites across the domain which together will make it very hard to see high-resolution spatial variations. To estimate these, we would rather test the capacity of satellite observations in combination with a gridded state vector, which is actually part of ongoing work in our group. In this first exploration, we however use the same spatial distribution for our pseudo-observations and for our prior.

We have added extra information explaining why we did not pay more attention to spatial downscaling and how general proxy data can be used in lines 339-344. We have also added a sentence explaining our assumption on not including spatial uncertainties in lines 359-363 and that this should be part of future work (lines 671-673).

*In addition to these major comments, I list several doubts and minor comments mostly related to suggestions to improve the presentation of the work:*

*L94: Change (Andres et al., 2016) (Super et al., 2019) for (Andres et al., 2016; Super et al., 2019)*

Done.

*L104: I think that the concept of "near real-time" is too strong. For instance, this would imply that traffic emissions are estimated based on near-real time data collected from traffic counts and, therefore, that congestion situations or traffic accidents are considered when calculating the dynamic emissions. A similar thing would apply to powerplants (e.g. emissions are derived from near-real time collected data on the activity of each individual facility).*

We agree with the reviewer that this concept is not applicable to the emission model shown here, but that it reflects what we would like to have in the future. We have updated line 104 and added some words about the future dynamic emission model in lines 155-159.

*L120: Replace "inverse part" for "inverse modelling part"*

Done.

*Table 1: Could you provide a reference to the CO2 contribution shares that are shown in Table 1?*

The source of this data has been added to the table caption.

*L206: Some European studies have suggested the use of 15.5°C as the value for defining the threshold temperature when calculating the HDD (e.g.https://rmets.onlinelibrary.wiley.com/doi/epdf/10.1002/joc.3959). According to the results shown in Figure 3 (left), the parametrization proposed for households (18°C) is underestimating most of the observed peaks in winter, while it overestimates the ones observed during spring/summer. On the contrary, the parametrization proposed for glasshouse (15°C) reproduces much better the winter peaks. Do you think that reducing the value of Tb for the household parametrization could allow improving the reproduction of winter peaks? (this is just a suggestion, does not need to be added in the revised manuscript)*

We appreciate the authors suggestions on this topic. Indeed, we are aware that some studies have used a different temperature threshold. We have adopted the value suggested by Mues et al. (2014), because they applied their method to Germany, which has a similar climatology as the Netherlands. To test the sensitivity, we compared the results using 18 and 15.5°C and the winter peaks are slightly better when using 15.5°C, yet the correlation coefficient remains the same. We did not include this in the manuscript.

*L220: Are you referring to the MACC-III fixed temporal profile? Please specify*

Yes, we used the diurnal profile from TNO-MACC. A reference is added to line 227.

*L239:Add a reference to the ENTSO-E database (e.g.https://www.sciencedirect.com/science/article/pii/S0306261918306068)*

Done.

*L244: The correlations presented between power generation and meteorological variables are rather low. This implies that the proposed parametrization for this sector is not so well correlated with observed activity data such as it is for other sectors (e.g .households or road transport). Considering the importance of this sector to the total*

*CO2 emissions, perhaps it would be interesting to discuss how these parametrizations could be improved in future works.*

We value the reviewer's suggestion to pay more attention to the energy sector emission timing. Indeed, it is a combination of uncertainty and absolute importance that determines where the effort should go. We think that especially the timing of coal-fired power plant activity can be improved by introducing a seasonal variation in the constant offset based on economic activity (e.g. lower industrial activity during the summer holidays). In contrast, the power generation from gas-fired power plants is more used as back-up for renewable energy. Yet, since the electricity supply is not local, the size of our domain limits correct calculation of the wind/solar shortage that needs to be supplemented by gas-fired power plants. These suggestions have been added to lines 267-273. How large the area should be to model energy supply is an interesting question for future research.

*L260: Could you also provide the R2 value for daily data?*

The $R^2$ values of coal- and gas-fired power plants (0.17 and 0.31) are added to lines 265-266.

*Table 1 / L272 / Figure 9: The industrial sector is the largest contributor to total CO2, but at the same time is the only sector that has not been split between subcategories. Is there a specific reason for that? Should not a split between e.g. type of industries would help to provide better temporal parametrizations or reduce the uncertainty of the emission factors for this sector?*

We thank the reviewer for pointing this out. Indeed, the industrial sector is an important source of $CO_2$ and in our study a major source of uncertainty. The large uncertainty has to do with the wide range of activities that are part of this sector that are difficult to capture in one emission factor, but also because it makes a lot of difference whether filtering technologies are implemented or not. Local knowledge can improve the emission factor estimate and for the case study region the emission factor is actually much better known. Splitting the industry up into subsectors could help reduce the uncertainty, but only if more specific information is available for each of the subsectors, which is often lacking or still includes a large uncertainty. Moreover, a further split in subsectors would add more parameters to our state vector, which are difficult to separate because industrial activities are often clustered in space. Therefore, we decided for now it would not be helpful to split this up into subsectors, as this would not reduce the overall uncertainty. We have added a statement on this in lines 145-147.

*Figure 5 (left): It looks like the activity data (red line) is missing for the last day*

We thank the reviewer for noticing the missing data. Unfortunately, the traffic count data are not complete and there is a small gap in the dataset. This is also why the sample size (line 308) differs and is smaller than a full year. This is now also mentioned in line 308.

*Figure 9: According to this figure, the uncertainty of the time profile "T" is larger in the household sector than in the power plant sector. Nevertheless, the correlation between the temporal parametrization and true activity data reported for the household sector is higher than the one reported for the power plant sector. Is there a specific explanation for that?*

We apologize for this confusion. The uncertainty in the temporal profiles is actually similar for the gas-fired power plants and households and somewhat lower for coal-fired power plants (based on the comparison between the

parameterizations and TNO-MACC fixed profiles, as mentioned in the footnote of Table A1). However, what is shown in Figure 9 is their contribution to the uncertainty in the emissions of that specific sector. So if another parameter is highly uncertain it will dominate the total emission uncertainty: the emission factor of coal-fired power plants has a larger uncertainty than the emission factor of households and therefore the relative importance of the uncertainty in the time profile is larger for households.

*Section 3.1: I assumed that the meteorological-dependent time profiles were calculated using the WRF model, but perhaps it should be clarified at some point in this section.*

We used the same meteorological data as for the calculation of the degree day sum as mentioned in the footnote of Table A1: observations from the Royal Netherlands Meteorological Institute (KNMI). This is now clarified in lines 222-223.

*Section 5: In the introduction section the authors pose three research questions that want to answer with this study. It would be interesting to rewrite the conclusions section so that it provide concise and clear statements that directly answers each one of these research questions (i.e. include a bullet list with a statement per question). This structure may facilitate the reader to link the posed questions with the outcomes of the work.*

We have carefully addressed this comment by rewriting the conclusion so that a clear and concise answer is provided for each research question (lines 787-796).

**Anonymous Referee #2

*The paper describes a new modelling framework to describe urban fossil fuel emissions of CO2 (ffCO2) in which emission ratios vary in time in space. To achieve this, the authors use atmospheric gases that are co-emitted with ffCO2 and range of proxy data that are associated with typical sectors that lie within the urban domain. They apply the resulting framework to a synthetic numerical experiment focused on the Rijnmond area, Netherlands.*

*This is a nice piece of work that with some development will eventually address some of the outstanding challenges we face as a community to quantify urban ffCO2. My recommendation is to accept the manuscript for publication after the authors have addressed my comments.*

*Broad comments*

*This is a chunky piece of work that contains a lot of information. For the sake of readability I encourage the authors to consider judicious use of additional appendices.*

We have carefully considered the reviewer's perspective and decided to create two additional appendices to reduce the amount of redundant details in the main text. Appendix B gives a summary of the data used to create the time profiles. Also the detailed explanation of the degree day function with corresponding equations is moved to Appendix B. Some of the details on the observation operator, which are not directly essential to the main point of the manuscript, have been moved to Appendix C.

*I have seen the authors present this work before and the use of "dynamic" has always rankled me. They could have just as easily described their new inventory as an online model that is fed with time-dependent data with*

*resulting emissions being passed directly to subsequent atmospheric calculations. This is in contrast with static or offline inventories. Static inventories can also be dynamic in time and space, albeit on a discrete basis.*

We appreciate the reviewer's thoughts on this and have carefully discussed it with all co-authors. We agree with the reviewer that what we present in the manuscript is not yet a full dynamic emission model. However, our ultimate goal is to develop a system that aggregates high-resolution activity data (traffic data, energy demand, shipping tracks) as well as the highly dynamic meteorological drivers of these activities. We moreover aim to access these in near real-time to calculate emissions for that specific moment only. We consider this approach to be justly called "dynamic" for several reasons: 1) it allows flexible use of different data sources including highly dynamic variables on activity/drivers that are not part of typical emission models or inventories; 2) it is not dependent on pre-calculated yearly emissions and spatial/temporal downscaling; 3) it provides emissions in near real-time. What we present here is a first step towards achieving this goal, namely by showing the potential of high-resolution activity data to describe temporal variations in emissions.

To address the reviewer's concern we have added a few sentences on what we believe a dynamic emission model should look like in lines 155-160 and the notion that what we present is just a first step towards achieving this.

*The figures are of low quality. Not sure why. I could barely read the text in Figure 1 and many of the other figures are grainy. Better quality figures will ultimately make the work easier to appreciate.*

*Figures would also benefit from being labeled directly, e.g A), B), C), etc. In some instances when columns are rows show something common a well-placed label would be useful. For example, Figure 4 would benefit from "Gas fired" and "Coal fired" labels for the top left and top right labels.*

We thank the reviewer for these suggestions. We have updated the figures accordingly.

*Bug bear: kindly please refrain from using "quite" as a descriptor throughout your pa-per. It is scientifically meaningless. Focus on the statistics that often accompany your statements.*

We have removed/replaced this term throughout the manuscript.

*Specific comments*

*Line 141: reason for greenhouses would be welcome here. Please mention tomatoes later but introduce the usage here.*

We have given more detail on the glasshouses in lines 141-142.

*Section 2.2.1. I think there might be a problem with units in your equation. Flux Fx should be mass/time but units of the contributing variables don't result in that unit. Please clarify units for all variables shown in equation 1.*

We thank the reviewer for pointing out this lack of clarity. The units are dependent on what type of activity data are used. In the case of GDP the unit of $A$ would be € (or another monetary unit), the unit of $E$ is PJ, the unit of $F$ is kg/yr and the unit of $R_x$ is kg/kg. Following equation 1 this would result in: € * (PJ/€) * ((kg/yr)/PJ) * (kg/kg) = kg/yr. So the units seem to be correct. We have added the missing units to lines 169-171.

*Figure 2. Please make this bigger.*

Done.

*Lines 203- 216 describe the definition of the time factor. I found the exposition of this point opaque, especially the accompanying mathematics. Please expound your argument.*

The usual approach for applying temporal disaggregation is to determine the average hourly emission in a specific year (yearly emission / number of hours) and then weigh them for each hour within the year using an hourly time factor ($T_t$). Over a full year, the average value of this factor is 1.0. This has been explained in lines 201-202. The degree day method has been used to calculate these hourly time factors for household emissions, which makes the timing dependent on the outside temperature. Basically, this method weighs all daily mean temperatures above a certain threshold and assigns emissions to these days accordingly, except for the constant offset that is equally spread over all days. A simplified explanation has been added to lines 217-220 and the details have been moved to Appendix B.

*Figure 3. The drop in relative gas consumption during May-Sept presumably reflects warmer weather. Are the spikes during this period due to cold days?*

Indeed, since the emission timing is purely dependent on the outside temperature those peaks reflect days on which the daily mean temperature exceeds the threshold. This explanation has been added to lines 237-238.

*Pages 8-9 I was unclear reading through this text how much was based fact, e.g. the reason behind gas-fired power plants (weakly) negatively correlated with wind speed, and how much was interpretation. Please clarify.*

We thank the reviewer for point out that our reasoning requires further explanation. The temperature-dependency of household/glasshouse emissions has been examined in detail and supported by observations. The values for the temperature threshold and constant offset are based on literature (households) or data fitting (glasshouses). This has been mentioned in lines 218-220 and 236-237.

As for power plants, the values have been chosen based on data fitting. The meteorological parameters are chosen based on a linear regression analysis with different types of meteorological data, from which we choose the parameters with the most explaining power (lines 250-251 and 254-255). In the case of gas-fired power plants this turned out to be wind speed and incoming solar radiation, which are predictors for the amount of wind and solar renewable energy. Since wind and solar energy are not always available and gas-fired power plants mainly serve as back-up during peak hours, we suggested that both support coal-fired power plants and that the energy mix depends on the amount of renewables that is available. This is our interpretation, which is now mentioned explicitly in line 255-256.

*Generally, this reader would appreciate a summary table that explains which variables are being used as proxy data for various urban sector emissions.*

A table has been made that summarizes the data used to create and validate the temporal profiles. It has been added to Appendix B.

*Curiosity: are gas-fired power plants quicker to respond to shortfalls in energy provision than coal-fired plants? Does this explain the weaker correlation reported in lines 259-261?*

We thank the reviewer for this interesting question. Since the temperature threshold for coal-fired power plants is relatively high, the temporal variations are relatively insensitive to the temperature, especially during the winter. The choice for this is indeed based on the knowledge that coal-fired power plants operate relatively continuously and respond less to chances in the temperature. Gas-fired power plants operate more dynamically, as they respond to chances in renewable energy supply (which are weather dependent). Moreover, an additional explanation for the lower correlation is due to our assumption that the offset is fixed throughout the year. We have tried to put this in perspective in lines 267-273.

*Uncertainty analysis shown in Section 2.4.1. is important for inverse modellers. Is this a stop-gap approach or do you envisage this as a final method?*

The methodology described in Section 2.1.4 is a first step towards a better quantification of parameters uncertainties and error correlations. We believe it is a promising method that, with some additional effort, could provide a flexible tool for inverse modelers. The main advantage is that it can include spatial uncertainties and therefore it can be applied irrespective of the required spatial/temporal resolution. A similar approach, albeit more detailed and including spatial uncertainties, has been recently published.

We have added some notions on the use of the uncertainty analysis in lines 373-375.

*Section 2.2. Convention dictates that vectors and matrices are denoted as emboldened lower- and upper-case variables, respectively.*

This has been updated throughout the text.

*Section 2.2.1. There is a lot being described here. Worth a schematic?*

A part of this section is moved to Appendix C, because it is of limited importance for interpreting the results. The most important features of the observation operator are summarized in Figure 7.

*Section 2.2.3. Closed-loop numerical experiments are considered useful only if the truth and prior are independent in some way. Some calculations might use independent inventories while others use independent transport models. Using the "dynamic" version of the static inventory is not sufficiently independent (e.g. Figures 5 and 6). Consequently, the authors have presented a very optimistic scenario. At least, the author should acknowledge this situation.*

We agree with the reviewer that the prior and truth are not completely independent. While the emission calculations use different, independent values, the spatial and temporal patterns are the same for the prior and truth in the experiments shown. Also the same atmospheric transport is used - except for the boundary conditions - although model errors are taken into account to estimate the observation error. However, we discussed experiments in which the temporal profiles and atmospheric transport are different for the prior and truth. In both experiments structural errors become the dominant and limiting factor, as supported by previous studies. While that setup represents a more realistic scenario, we believe it doesn't support our goal to explore the potential of an inversion system with a dynamic fossil fuel emission model and co-emitted species. We have added a statement on this in lines 481-484.

*Section 2.2.3. The authors assume no contribution from biogenic CO2 to the excess CO2 over the background. This is not a general assumption. How will they cope with an urban area with parks, for example?*

The biogenic fluxes are included in the background mole fractions of CarbonTracker Europe (lines 459-460). What we assume is that the biogenic contribution is the same in the prior and true background, so that the error in the background/biogenic flux is attributed to the fossil fuel emissions. This represents a typical situation in which the fossil fuel signal is difficult to isolate from the total mole fraction. The presence of biogenic fluxes thus contributes to the uncertainty in the fossil fuel flux estimates, which makes this exercise more realistic. In a study using real observations biogenic fluxes can be treated in more detail, e.g. with a biogenic emission model, and an effort can be made to separate the fossil and biogenic signal with isotopes. This explanation has been added to lines 499-502.

*Section 2.2.3. A few more details are necessary to describe the data. Ideally, earlier in the manuscript. I am surprised that the authors can achieve what they have with a handful of data collected at 10 metres a.s.l. Maybe this can only work in the Netherlands? Also, what is the origin of the values used in the R matrix?*

We only consider 7 observation sites with an inlet height of 10 m and select observations between 12 and 16h LT (lines 503-509). Nevertheless, we do have 4 species, which all add information to the inversion system. With this setup we have a total of 1930 observations to constrain 44 parameters as mentioned in lines 406 and 506. This is an important advantage of using co-emitted species. Normally using in-city ground-level observations can be challenging due to erroneous transport, but since we use the same transport for the truth and the optimization this is not an issue. This notion is added to lines 509-511. Some more detail on the R matrix is given in lines 517-518.

*Section 3. State that CI = confidence interval. Also, clarify "Below the annual scale" online 543.*

The meaning of CI has been given in line 379. "Below the annual scale" has been replaced with "At the sub-annual time scale" in line 554.

*Section 3.2. The result associated with a shortened state vector was interesting and something this reviewer had not considered fully. How do we decide on the correct length of the state vector? Will this be location specific?*

This is a very interesting question and difficult to answer. Based on our results we believe that parameters with minor influence on the total emissions (e.g. because the contribution of the source sector is small) can still be important if they are highly uncertain. However, including all these minor parameters will make the state vector very large and introduce more correlations which hamper the separation between parameters. Therefore, we can imagine that in this example the $CO_2$ mole fractions might get more weight than the mole fractions of co-emitted species if the emission ratios of those species are very uncertain. However, how to best approach this is worth a further examination. We have added a sentence on our interest to further explore this in lines 665-667.

*Minor comment: avoid using yellow in figures (Figure 10).*

The yellow lines have been replaced by green dashed lines to increase the readability of the figure.

*Figure 11 would benefit from a legend. It contains a lot of information that was all in the text and figure panel but it took a while to pick through it all.*

A legend has been added to the figure.

*Line 650. I would say that this approach provides a more detailed physical meaning of the results compared to estimating emission estimates.*

We thank the reviewer for this nuance and agree that emissions in itself are also important physical results. Line 662 have been updated.

*Line 652. Non-included parameters?*

We refer to those parameters that have an uncertainty, but are not part of the state vector and therefore not optimized. This has been clarified in lines 663-664.

*Line 659. If your online inventory is using weather data to drive variations then you could use the correlation lengths associated with weather systems?*

We thank the reviewer for this suggestion. Indeed, a correlation length based on typical weather system characteristics could be helpful to determine over which time period data are correlated. However, this would only apply to certain source sectors which depend on weather conditions, so other approaches are needed to determine typical correlations lengths for other sectors. We have added the reviewer's suggestion as an example in lines 678-680.

*Section 4.2. Putting all your eggs in one basket with radiocarbon is not a wise move. It is one weapon in your arsenal. With the growth of biofuel combustion in urban regions, there will be a lot of combustion CO2 that is missed using radiocarbon. Something to consider in your discussion, especially since your group has just published work on this topic that makes my point.*

We agree with the reviewer that radiocarbon is not the solution to all problems. However, it is mentioned here as it has been proven useful in several inverse modelling studies. Since our emission model only contains fossil fuel emissions, radiocarbon is definitely a good addition to constrain the model parameters. Nevertheless, we agree that there are several other fluxes that need to be considered, such as biogenic and biofuel combustion fluxes. Therefore, we have mentioned oxygen ($O_2$ oxidative ratios), as well as several other isotopes that can be used to distinguish between different fuel types (lines 700-703) and the option to optimize the background (lines 704-715). Ideally, a combination of these techniques is applied to gain as much information as possible on the distinct sources of $CO_2$.

*Line 774. Are you saying that your model has an advantage because it uses a source of information (emission-related parameters) that is often neglected by emission inventories?*

Regular emission inventories often calculate emissions based on energy consumption statistics in combination with emission factors, which is not much different from our approach. The main difference is that emission maps do not explicitly contain the underlying data anymore and therefore do not allow to optimize those parameters. Another advantage is that, because it is a model and not a static map, it can use local high-resolution data when available. Therefore, it is more flexible than a regular emission inventory and the calculation of its uncertainties is substantially easier and more transparent. We have clarified this in lines 788-796.

---

## Author Response (AR2)

***Topical editor comment on "Optimizing a dynamic fossil fuel CO2 emission model with CTDAS (v1.0) for an urban area using atmospheric observations of CO2, CO, NOx, and SO2" by Ingrid Super et al.***

*Thank you for the thorough response to the reviewers. I have gone them through and have only minor technical suggestions that should be addressed:*

We would like to thank the topical editor for reviewing our revised manuscript and for providing additional feedback to improve our manuscript.

*L172: Should be (Fco2/E) and not (F/E) corresponding the equation itself*

We have corrected this to ensure consistency between the text and the equation.

*L183-185: It is not clear from the additional text to which "These data" on line 183 refers to. My understanding is that the sentence concerns A(E/A) part of the equation and not only A. Thus I would consider how to clarify the text here on L181-189.*

We thank the editor for pointing out this ambiguity. Actually, we do refer to *A*, which can be based on widely available statistics like GDP as was discussed in previous lines. This has been clarified.

*L341: I think there should be the in front of absence-> the absence*

What we mean to say here is that if there is no high-resolution inventory, proxies can be used for spatial disaggregation. So we believe that the sentence is correct like this.

*L360: The 7 (pseudo-)observation stations are mentioned now here for the first time as they are explained only in Section x.x. Thus it would be good to specify here something like 7 stations in where the approach will be evaluated.*

We thank the editor for pointing this out. We have updated the sentence and added a reference to the section with more details on the observation network.

*L501: Should be "fluxes can be significant"*

This has been updated.

*You write currently in the Code and data availability section that "The source code available from https://git.wageningenur.nl/ctdas/CTDAS forms the basis of the system described in this paper.". References to webpages should be avoided as they are not permanent (not even in git). As the full code is provided as an supplement I suggest to remove the whole sentence as you have the reference included on the previous sentence.*

The reference to the webpage has been removed.

[revised manuscript text omitted]